



# Sensitivity of winter Arctic amplification in NorESM2

Lise Seland Graff[1], Jerry Tjiputra[2,4], Ada Gjermundsen[1], Andreas Born[3,4], Jens Boldingh Debernard[1], Heiko Goelzer[2,4], Yan-Chun He[5,4], Petra Margaretha Langebroek[2,4], Aleksi Nummelin[2,6], Dirk Olivié[1], Øyvind Seland[1], Trude Storelvmo[7], Mats Bentsen[2,4], Chuncheng Guo[2,8], Andrea Rosendahl[1,3], Dandan Tao[3,4], Thomas Toniazzo[2,4], Camille Li[3,4], Stephen Outten[5,4], and Michael Schulz[1,7]

[1]Norwegian Meteorological Institute, Oslo, Norway
[2]NORCE Norwegian Research Centre AS, Bergen, Norway
[3]University of Bergen, Bergen, Norway
[4]Bjerknes Centre for Climate Research, Bergen, Norway
[5]Nansen Environmental and Remote Sensing Center, Bergen, Norway
[6]Finnish Meteorological Institute, Helsinki, Finland
[7]University of Oslo, Oslo, Norway
[8]National Centre for Climate Research, Danish Meteorological Institute, Copenhagen, Denmark

**Correspondence:** Lise Seland Graff (lise.s.graff@met.no)

**Abstract.**

   While Arctic amplification is a robust feature of both observed and projected climate change, projections of Arctic climate change are characterized by substantial uncertainty. To better understand the drivers of this uncertainty, we performed a coordinated set of fully coupled experiments with the second version of the Norwegian Earth System Model (NorESM2) in which

5  selected processes of key importance for the Arctic climate have been modified. They include improved representation of (1) mixed-phase clouds, (2) eddy processes in the upper ocean, (3) Greenland ice-sheet coupling, (4) snow on sea ice processes, and (5) ozone chemistry. For each modification, we carried out sensitivity experiments following the protocols for the CMIP6 historical simulation and a future high-emissions scenario (ssp585). This results in an ensemble of modified historical and ssp585 experiments.

10  The sensitivity experiments all demonstrate enhanced future Arctic warming compared to the unmodified historical and ssp585 experiments. The amplitude of the additional warming moreover varies considerably, with the difference between the experiment with the strongest and weakest Arctic-mean warming reaching ∼9 K during the winter season by the end of the 21st century. The warming signal is dominated by a relatively uniform Arctic warming which, according to the CMIP6 ssp585 long-term extension, starts to equilibrate during the 22nd century. Surface temperature decomposition shows that winter warming

15  is primarily driven by enhanced greenhouse effect due to increased cloud cover, near-surface humidity, and the resulting increase in downwelling longwave radiation. The temperature response is most pronounced in the sea-ice retreat regions, with the greatest variability between experiments occurring on the Atlantic side. We also identify an emergent constraint, linking changes in Arctic surface temperatures to changes in ocean heat fluxes and sea-ice area. This highlights the importance of correctly representing (contemporary) Northern Hemisphere sea ice when assessing future projected Arctic warming.



## 1 Introduction

The surface warming that occurs in response to changing conditions in the Earth system, such as increased concentrations of greenhouse gases (GHGs), is not uniformly distributed throughout the atmosphere, but is amplified at upper levels in the tropics and at lower levels in the Arctic (e.g., Lee et al., 2021). Since the late 20th century, the Arctic has been warming faster than the global mean (e.g., Cohen et al., 2020). This phenomenon is commonly known as Arctic amplification, and is a characteristic feature of both observed (e.g., Serreze et al., 2009; Bekryaev et al., 2010) and projected climate change (e.g., Forster et al., 2021; Davy and Outten, 2020; Graff et al., 2019).

Arctic amplification was identified more than a century ago (Arrhenius, 1896) and has since been attributed to several key processes. A well-known contributor is the loss of sea ice and snow (e.g., Screen and Simmonds, 2010; Serreze et al., 2009; Previdi et al., 2021), which are associated with a positive ice-albedo feedback and an increase in the heat fluxes from the ocean to the atmosphere during winter. Other key processes include increased poleward transport of heat both in the atmosphere and ocean, and the non-uniform warming of the troposphere, with stronger warming at lower levels in the Arctic enhancing surface warming through the lapse-rate feedback (e.g., Pithan and Mauritsen, 2014; Feldl et al., 2020; Previdi et al., 2021).

Despite our advanced understanding of relevant processes and driving mechanisms of Arctic amplification, simulations from the sixth phase of the Coupled Model Intercomparison Project (CMIP6; Eyring et al., 2016) reveal that projected future warming is characterized by larger uncertainty in the Arctic than in any other region on the planet (Taylor et al., 2022). Due to the broad diversity in physical components and respective process and feedback representations in Earth System Models (ESMs), it is not straightforward to elucidate the main driver of this uncertainty. The uncertainty in the projected Arctic climate change further translates to a large spread in the projected environmental impacts such as marine primary production (Myksvoll et al., 2023). Based on previous studies, we identify five processes across different Earth system components that could contribute to the uncertainty in the projected surface energy budget in the Arctic. They are cloud microphysics (e.g., Tan and Storelvmo, 2019; Morrison et al., 2012), ocean eddy parametrizations (e.g., Li et al., 2024; Oldenburg et al., 2024; Manucharyan and Thompson, 2022), Greenland ice-sheet coupling (e.g., Vizcaíno et al., 2010; Goelzer et al., 2011; Muntjewerf et al., 2020), snow on sea ice (e.g., Holland et al., 2021), and atmospheric chemistry (e.g., Gettelman et al., 2019; Orbe et al., 2024). To investigate the potential impacts of these processes, we perform a coordinated ensemble of sensitivity simulations using a single CMIP6 ESM.

We use the second version of the Norwegian Earth System Model (NorESM2; Seland et al., 2020b) with the following modifications: (1) *improved representation of mixed-phase clouds*, achieved by rectifying an error in the ice-crystal nucleation (Shaw et al., 2022; McGraw et al., 2023) and biases in the representation of cloud phase (Cesana et al., 2015); (2) *improved eddy processes in the upper ocean*, achieved by enhancing the representation of continental boundary currents and ocean fronts (Nummelin and Isachsen, 2024); (3) *updated representation of the Greenland ice-sheet coupling*, achieved by replacing the prescribed ice-sheet geometry and hydrological cycle with a dynamically changing ice-sheet model (Goelzer et al., 2025; Haubner et al., 2025); (4) *improved snow (on sea ice) processes*, achieved by adjusting parameters in the snow scheme to correct biases in sea-ice extent and volume (Seland et al., 2020b); and (5) *interactive ozone chemistry*, achieved by enhancing





the complexity of the atmospheric model chemistry allowing for explicitly describing ozone instead of prescribing it from an
external climatology file.

For each modification, we have completed a historical experiment in accordance with the CMIP6 protocol (Eyring et al., 2016), and a future scenario experiment, following the protocol for the Shared Socioeconomic Pathway corresponding to increased radiative forcing of 8.5 W m$^{-2}$ by the end of the 21st century (ssp585; O'Neill et al., 2016). The sensitivity experiments are as similar to the CMIP6 historical and ssp585 baseline experiments as possible, differing only in the specific model modifications listed above. We refer to this set of of experiments as the NorESM2 Ensemble Exploring Model Sensitivity (NEEMS). To assess the influence of the updates, we compare the results from the historical and ssp585-based NEEMS experiments to results from the baseline historical and ssp585 experiments carried out with NorESM2 for CMIP6 (Bentsen et al., 2019a, e). To put the results into context, we additionally compare with NorESM2 results from a range of scenarios from the Scenario Model Intercomparison Project (ScenarioMIP; O'Neill et al., 2016), and with experiments without anthropogenic aerosols.

Results show that all sensitivity experiments are characterized by enhanced future surface warming in the Arctic by the end of the 21st century compared to the warming in the baseline CMIP6 experiments (ssp585−historical). The end-of-the-century range of warming across the sensitivity experiments is moreover comparable to the ScenarioMIP warming range, defined as the difference between ssp585 and the SSP corresponding to increased radiative forcing of 2.6 W m$^{-2}$ (ssp126; O'Neill et al., 2016). To identify and better understand the key drivers of this large warming range, we decompose the change in surface temperature following Lu and Cai (2009) and Boeke and Taylor (2018), and explore the most important processes associated with the dominant drivers.

In what follows, we first provide an overview of NorESM2 and the experiments in Sect. 2 and 3. We then present the results in Sect. 4 and summarize and discuss our findings in Sect. 5. Additional figures are provided in the supplementary material.

## 2 Model

NorESM2 is based on the Community Earth System Model version 2 (CESM2), sharing the model infrastructure and several model components. The ocean component, the Bergen Layered Ocean Model (BLOM; Seland et al., 2020b), and ocean biogeochemistry component, the isopycnic coordinate Hamburg Ocean Carbon Cycle model (iHAMOCC; Tjiputra et al., 2020), are however entirely different. The atmospheric component, a modified version of the Community Atmosphere Model version 6 (CAM6-Nor; Seland et al., 2020b), is based on the CAM6 version used in CESM2 (Danabasoglu et al., 2020), but has a different module for aerosol physics and chemistry, and includes updates in the representation of dry and moist energy conversion, local and global angular momentum conversion, deep convection, and air-sea fluxes. There are also some minor changes in the land component, the Community Land Model version 5 (CLM5; Lawrence et al., 2019), and sea-ice component, the Community Ice CodE version 5 (CICE5; Hunke et al., 2015). The CMIP6 version of NorESM2 does not include dynamically changing ice sheets, and both the Greenland and Antarctic ice sheets are fixed to their three-dimensional present-day geometry.

We use the medium-resolution version, NorESM2-MM, which has nominal $1° \times 1°$ resolution in the horizontal in the atmosphere, land, and ocean, 32 hybrid-pressure layers in the atmosphere, and 53 layers in the ocean component. While not used



**Table 1.** Experiment overview. The experiment name is provided in the first column, a brief description in the second column, and the experiment type ("baseline", "NEEMS", or "other") in the third column. The *baseline* experiments are the historical and ssp585 experiments from CMIP6 that the NEEMS experiments are based on; the *NEEMS* experiments are the historical- and ssp585-based sensitivity experiments with various model improvements implemented; and the *other* experiments are other CMIP6 or CMIP6-based experiments used for comparison.

| Experiment | Description | Type |
| --- | --- | --- |
| historical | CMIP6 historical | Baseline |
| hist-cloud | As historical, but with improved mixed-phase cloud processes | NEEMS |
| hist-eddy | As historical, but with improved eddy processes in the upper ocean | NEEMS |
| hist-iceSheet | As historical, but with an active Greenland ice-sheet model | NEEMS |
| hist-snow | As historical, but with improved snow (on sea ice) processes | NEEMS |
| hist-ozone | As historical, but with interactive ozone chemistry | NEEMS |
| hist-piAerOxid | As historical, but without anthropogenic aerosols | Other |
| ssp585 | CMIP6 ssp585 | Baseline |
| ssp585-cloud | As ssp585, but with improved mixed-phase cloud processes | NEEMS |
| ssp585-eddy | As ssp585, but with improved eddy processes in the upper ocean | NEEMS |
| ssp585-iceSheet | As ssp585, but with an active Greenland ice-sheet model | NEEMS |
| ssp585-snow | As ssp585, but with improved snow (on sea ice) processes | NEEMS |
| ssp585-ozone | As ssp585, but with interactive ozone chemistry | NEEMS |
| ssp585-piAerOxid | As ssp585, but without anthropogenic aerosols | Other |
| ssp370 | CMIP6 ssp370 | Other |
| ssp245 | CMIP6 ssp245 | Other |
| ssp126 | CMIP6 ssp126 | Other |

here, the low-resolution version, NorESM2-LM, which has a nominal $2° \times 2°$ horizontal resolution in the atmosphere and land models, will be referred to in the text. For a more comprehensive model description and evaluation, see Seland et al. (2020b).

## 3 Experiments

90 We categorize our experiments into three groups: (1) baseline experiments, which are the NorESM2-MM historical and ssp585 experiments carried out for CMIP6; (2) NEEMS experiments, which, similar to the baseline experiments, follow the protocols for the CMIP6 historical (Eyring et al., 2016) and ssp585 (O'Neill et al., 2016), but include different model updates implemented individually; (3) other experiments used to put future changes induced by the model updates into context. For the NEEMS experiments, a historical and ssp585-based experiment set is completed for each model update (*cloud*, *eddy*, *iceSheet*,

95 *ozone*, and *snow*; note that we use italics when referring to experiment names without the hist- or ssp585- prefixes for clarity). In line with the relevant CMIP6 protocols, all historical experiments cover the time period 1850–2014 and all ssp585-based





experiments cover years 2014–2100. The baseline and *iceSheet* versions of ssp585 are extended up to year 2300. While we focus primarily on the difference between the last 30 years of the 21st century (2071–2100) and the last 30 years of the historical (1985–2014), we consider the full time series (1850–2100) in Sect. 4.1 and 4.3 and the full extension (1850–2300, available for the baseline ssp585 and ssp585-iceSheet only) in Sect. 4.4.

An overview of the experiments is provided in Table 1 and further details are given below. For the *cloud*, *eddy*, and *iceSheet* experiments, the model updates have been previously documented in other papers and hence we only provide a summary and refer to the relevant papers for further details. For the *snow* and *ozone* experiments, we provide a more comprehensive description as the experiments are first documented in the present paper.

## 3.1 The *cloud* experiments

In the *cloud* experiments (hist-cloud and ssp585-cloud), we introduce changes to the NorESM2 cloud microphysics scheme that (1) improve the model's ability to produce changes in the ice crystal number due to heterogeneous ice nucleation (Shaw et al., 2022; McGraw et al., 2023), (2) address a well-known ESM bias in cloud thermodynamic phase relative to satellite observation (Cesana et al., 2015), and (3) address an imbalanced energy budget at the top-of-the-atmosphere, as a consequence of addressing (1) and (2).

In NorESM2, heterogeneous ice-crystal nucleation (i.e., ice nucleation with the aid of ice-nucleating particles; INPs) is aerosol-dependent and follows the parameterization by Hoose et al. (2010). However, an overlooked limiter, which sets the maximum number of cloud ice particles, inadvertently negated this scheme's ability to produce changes in the ice crystal number due to the heterogeneous ice nucleation. This limiter has been replaced in *cloud* by one that simply ensures that the number of nucleated ice crystals does not exceed the number of available INPs (as originally intended), following Shaw et al. (2022) and McGraw et al. (2023).

However, with this correction alone, the fraction of cloud ice in the temperature range between $-40$ and $0°C$ increases considerably, and is no longer consistent with cloud phase observed by active remote sensing. This is problematic, because it is well established that cloud phase exerts a strong influence on the simulated extratropical cloud feedback and thus climate sensitivity (Tan et al., 2016). This issue is addressed in *cloud* by also reducing the efficiency of the Wegener-Bergeron-Findeisen process (i.e., the conversion from liquid to ice in mixed-phase conditions), adjusting the fraction of dust and black-carbon aerosols that are assumed to act as INPs, and changing the assumed thermodynamic phase of convective detrainment, such that cloud phase matches satellite observations both qualitatively and quantitatively (Hofer et al., 2024).

As a consequence of the changes described above, the model no longer had a balanced top-of-the-atmosphere radiation budget, and required tuning before a plausible model state in the pre-industrial control simulation could be established. The retuning was done by changing one of the main tuning parameters in CESM, "Clubb_gamma_coef" which impacts the entrainment at the top of the planetary boundary layer and in turn changes the low stratiform cloud cover (Seland et al., 2020b; Danabasoglu et al., 2020). To remedy the negative radiative imbalance at the top of the atmosphere, the parameter was increased compared to the CMIP6-version of NorESM2-MM from 0.286 to 0.330.





The historical experiment with the *cloud* changes (hist-cloud) started directly from the end of the spin-up experiment that was carried out for the CMIP6 baseline experiments (hereafter referred to as the baseline spin-up), which is a well-balanced state with negligible drift (Seland et al., 2020b). The scenario (ssp585-cloud) started from the end of the hist-cloud.

### 3.2   The *eddy* experiments

In the *eddy* experiments (hist-eddy and ssp585-eddy), we follow Nummelin and Isachsen (2024) and improve the representation
of continental boundary currents and ocean fronts by making the mesoscale eddy parameterization bottom-slope aware, an effect which is particularly important at high latitudes.

The state-of-the-art ESMs used in long climate integrations generally do not resolve the mesoscale eddy field in which most of the ocean's kinetic energy resides (Ferrari and Nikurashin, 2010). Consequently, the reduction of available potential energy and tracer gradients, which is the main effect of the mesoscale eddies, is parameterized, most often following the
works of Gent and McWilliams (1990, hereafter GM90) and Redi (1982). These parameterizations typically estimate an eddy transport (diffusion) coefficient using mixing length theory as the product of a velocity scale and a length scale. In estimating these scales, theory relies on idealized assumptions, for example, a flat ocean bottom and isotropic eddy-mixing. However, it is well understood that potential vorticity gradients asserted by sloping topography steer the flow and reduce cross-gradient mixing, and indeed, Sterl et al. (2024) recently showed that based on barotropic potential vorticity dynamics, the eddy mixing
coefficient is expected to have squared dependence on bottom slopes.

To account for such a reduced diffusivity over sloping bottom topography, we follow Nummelin and Isachsen (2024) and introduce a topographic Rhines scale (Rhines, 1977) as an eddy length scale. In addition, we include a diagnostic eddy efficiency factor that further scales down the diffusivity when the parameterized eddy velocity scale remains small despite large potential for baroclinic instability (i.e., strong thermal wind; Nummelin and Isachsen, 2024). These modifications are expected
to reduce the eddy diffusivity over sloping topography at the middle and high latitudes, to strengthen the boundary and slope currents and to intensify fronts. Although most of the theory was developed for the eddy buoyancy transport, here, for simplicity, we apply the same coefficient in both the GM90 and Redi parameterizations – this choice was shown to lead to best results in ocean-only simulations (Nummelin and Isachsen, 2024, see their supplementary material). The implementation of the modified eddy parameterization and the corresponding idealized and ocean- and ice-only test cases are described in detail
in Nummelin and Isachsen (2024).

The fully coupled model experiments were started from the end of the baseline spin-up, and were then spun-up for another 100 years. During the spin-up, some parameters as implemented in Eden and Greatbatch (2008) for the GM90 diffusivity were tuned as follows compared to the CMIP6 setup to achieve a balanced top-of-the-atmosphere energy budget and Southern Ocean sea-ice extent: (1) the hyper-parameter $c$ was changed from 0.85 to 2, (2) the minimum diffusivity limit was changed
from $100.0 \times 10^4$ cm$^2$ s$^{-1}$ to $1.0 \times 10^4$ cm$^2$ s$^{-1}$, and (3) the maximum diffusivity limit was changed from $1500.0 \times 10^4$ m$^2$ s$^{-1}$ to $2500.0 \times 10^4$ m$^2$ s$^{-1}$. After the 100 year spin-up was completed, a pre-industrial control (piControl-eddy; not considered here) and a historical experiment (hist-eddy) were started from the end of the spin-up, and then the scenario experiment (ssp585-eddy) was started from the end of hist-eddy.





### 3.3 The *iceSheet* experiments

The purpose of the *iceSheet* experiments (hist-iceSheet and ssp585-iceSheet) is to introduce a dynamically changing Greenland ice sheet by including a new model component, the Community Ice Sheet Model (CISM; Lipscomb et al., 2019). The standard version of NorESM2, and hence the baseline experiments, represents the Greenland and Antarctic ice sheets in terms of fixed present-day geometries. In the *iceSheet* experiments, the Greenland ice-sheet geometry is dynamically changing in response to the climate forcing (Goelzer et al., 2025) following a similar procedure as in CESM2-CISM2 (Muntjewerf et al., 2020, 2021).

The Antarctic ice sheet remains fixed to its present-day geometry and is not changing with climate forcing.

The surface mass balance of the Greenland ice sheet is calculated within the land model based on an elevation class approach (Lipscomb et al., 2013; Sellevold et al., 2019) that downscales the surface energy and mass-balance components from the land-model grid at $1° \times 1°$ resolution to the $4\,\text{km} \times 4\,\text{km}$ grid of the ice-sheet model. Freshwater input to the ocean model from the ice sheet occurs as both meltwater runoff and iceberg calving. Fluxes between the ice-sheet model and the other components

are updated once a year, except for the topographic and roughness boundary conditions for the atmosphere that change with evolving ice-sheet geometry, which are updated every five years (Goelzer et al., 2025).

The *iceSheet* experiments used the same tuning as the CMIP6 baseline experiments, because the pre-industrial ice-sheet configuration is very close to the prescribed ice sheet in the baseline experiments. However, comparing the Greenland ice-sheet evolution with the multi-model ensemble of the Ice Sheet Model Intercomparison Project for CMIP6 (ISMIP6; Nowicki

et al., 2016) shows that the mass loss by 2100 in the *iceSheet* experiments is near the low end of the ISMIP6 range. This can in part be linked to a cold temperature bias in NorESM2 around the Greenland margin (c.f., Seland et al., 2020b), which suggests that additional tuning may be necessary to improve the ice-sheet simulations. However, such a tuning would not be limited to the ice-sheet component but impact the full model state, including land, the representation of Arctic amplification, and changes in the dynamics of meridional heat transport in both atmosphere and ocean. As this is beyond the scope of this study, we did

not perform such tuning here.

### 3.4 The *snow* experiments

The *snow* experiments were motivated by the fact that the two different resolutions of NorESM2 used in CMIP6 have very different sea-ice volume in the Arctic. While the low-resolution version NorESM2-LM has a reasonable representation of the present-day climate, the medium-resolution version NorESM2-MM has too much sea ice, both in extent and thickness (Notz

and Community, 2020). Due to the excessive sea-ice cover in the historical period, the MM-version is one of the CMIP6 models with the slowest decline in ice cover. One of the main differences between the two model versions is that the MM-version is almost 1 K colder during the piControl simulation, which we hypothesize to be the main cause of the sea-ice differences. In the CMIP6 experiments, the LM- and MM-versions shared identical sea-ice tuning and physics. The *snow* experiments aim to investigate how tuning, primarily of snow conductivity, might align the historical Arctic sea-ice cover in the MM-version more

closely with observations.



The thermal conductivity of snow varies strongly as a function of density, but also crystal structure and water content play important roles. Snow on Arctic sea ice in winter is dominated by dense layers of wind-packed snow, with relatively high conductivity, overlaying layers of faceted crystals or depth hoar (DH). Typically, around 50% of Arctic sea ice consists of DH or similar highly transformed layers (e.g., Merkouriadi et al., 2017). Especially for DH-layers, the spread in cited
conductivity values is considerable, both with respect to observation method, and physical conditions. While measurements based on needle-probe techniques often show low conductivity with little dependence on density (Sturm et al., 1997), estimates based on micro computer tomography combined with finite element modelling show higher values and more dependence on the density (Calonne et al., 2011; Macfarlane et al., 2023), the latter being more in line with temperature-gradient-based estimates (Calonne et al., 2011).

The discrepancy in conductivity measurements gives rise to a large range of snow conductivity estimates for the same snow pack. Based on observations from the SHEBA drift experiment, Sturm et al. (2002) defined a mean snow pack with different layers corresponding to the conditions observed in the snow pits, together with mean density, snow type, hardness, snow conductivity (needle-probe based), and thickness of the layer. Using the fact that the total heat resistance over the snow pack can be calculated as the sum of the resistance over all the individual layers (Macfarlane et al., 2023), the mean effective
conductivity can be defined as $K_H = (\sum(dh_i/k_{si})/\sum(dh_i))^{-1}$, where $k_{si}$ and $dh_i$ are the effective conductivity and thickness of each layer, respectively. Using the needle-probe based observations of conductivity from Sturm et al. (2002), this gives an effective conductivity of $0.165$ W m$^{-1}$ K$^{-1}$ for the snow pack, while the results from different density-based parameterizations give $0.27$ W m$^{-1}$ K$^{-1}$ (Yen, 1981), $0.15$ W m$^{-1}$ K$^{-1}$ (Sturm et al., 1997), $0.23$ W m$^{-1}$ K$^{-1}$ (Calonne et al., 2011), and $0.29$ W m$^{-1}$ K$^{-1}$ (Macfarlane et al., 2023).

Lecomte et al. (2013) formulated conductivity and density as functions of seasonal wind speed, and tested this in a coupled ice-ocean model in combination with the formulations of Yen (1981) and Sturm et al. (1997), but also in combination with a constant value of $0.3$ W m$^{-1}$ K$^{-1}$, which is similar to what was used in the NorESM2 CMIP6 experiments. They concluded that the formulation of Yen (1981) and the new wind-based formulation gave the best results in the Arctic, while the formulation of Sturm et al. (1997) looked best in the Antarctic, with their particular atmospheric forcing of the model. Monthly anomalies
in ice extent were also best described by the latter formulation. Based on recent measurements, Macfarlane et al. (2023) suggest that the modelling community should explore the effects of decreasing the bulk snow conductivity in model runs.

The heat conductivity in the *snow* experiment was reduced from $0.30$ W m$^{-1}$ K$^{-1}$ to $0.15$ W m$^{-1}$ K$^{-1}$, close to the values used in Fichefet et al. (2000) and Wu et al. (1999). This resulted in a considerable reduction in Arctic sea-ice volume. In the Antarctic, this large change causes a pronounced increase in snow-ice conversion. The new conductivity is most likely too low
when considering the large-scale sea-ice-snow system. But, as a sensitivity experiment, it should be valuable for exploring the coupled system response.

A spin-up run utilizing the new snow conductivity was initialized from the end of the baseline spin-up, and run for 100 years. A 250-year pre-industrial control (piControl-snow; not considered here) and the 165-year historic run (hist-snow) were both started from the end of the spin-up run and the 86-year scenario (ssp585-snow) from the end of hist-snow.





## 3.5 The *ozone* experiments

In the *ozone* experiments (hist-ozone and ssp585-ozone), we improve the representation of tropospheric and stratospheric chemistry by introducing a chemistry scheme that enables explicitly describing ozone and other chemical species rather than prescribing their concentrations from forcing files.

The changing atmospheric composition (due to presence of GHGs and aerosols) is one of the main drivers of climate change, due to its interaction with radiation and clouds. In the CMIP6 version of NorESM2, atmospheric GHGs and ozone distributions are prescribed, and only the distribution of aerosols is prognostically calculated in the model. The prognostic aerosol scheme consists of eight gas-phase tracers together with four oxidant climatologies, and 21 aerosol tracers (to describe the aerosol population consisting of black carbon, organic matter, sulphate, dust, and sea salt). The long-lived GHGs ($CO_2$, $CH_4$, $N_2O$, CFC-11, and CFC-12) are prescribed as time-varying surface concentrations combined with a species-specific vertical profile which varies with height and latitude. Ozone is prescribed as a monthly-varying climatological two-dimensional (zonally-symmetric) distribution. In the CMIP6 version of the model, the prescribed fields are derived from experiments with CESM2, where CAM6 is replaced by the Whole Atmosphere Community Climate Model (WACCM; Gettelman et al., 2019), yielding CESM2-WACCM, a high-top model with detailed atmospheric chemistry.

In the *ozone* experiments, we introduce tropospheric and stratospheric chemistry in NorESM2 by activating the TS1 chemistry scheme (available from CESM; Emmons et al., 2020), and coupling it to the NorESM2 aerosol scheme. This results in ozone and many other chemical species being explicitly described, and the number of explicitly modeled aerosol and gas-phase species increases from 29 to 210. The earlier-described oxidants and long-lived GHGs are now also explicitly represented in the chemistry scheme. The long-lived GHG concentrations at the surface are still prescribed, but their vertical profiles are determined by atmospheric transport, chemical reactions, and photolysis. The radiation code in the model sees these calculated three-dimensional distributions of ozone and long-lived GHGs.

In addition to introducing a more complete atmospheric chemistry description, we also have modified the dry-deposition of aerosols. This modification leads to increased lifetime of large aerosols (coarse dust and sea salt), which increases aerosol optical depth due to natural background aerosol by around 0.02.

There are no modifications to any tuning parameters in these experiments, although there is a small cooling trend under pre-industrial conditions (visible at the global scale in the top-of-the-atmosphere radiative balance, and in the total ocean heat content; not shown). We started the *ozone* experiments from the end of the baseline spin-up, and ran a spin-up simulation for 60 years with the updated version of the model. At that point, we started the historical (hist-ozone) and after that the scenario (ssp585-ozone) simulations.

## 3.6 Other experiments

Throughout this study, our focus is on how the various model updates described above affect future changes in the Arctic. However, to put these results into context, we will compare the future changes due to the model updates to future changes found in other warming scenarios and in experiments without anthropogenic aerosols. In addition to our future baseline experiment



**Table 2.** Definitions of the future response (sspRes) and future response difference (sspRes-diff) for the experiments. We use years 1985–2014 for all historical-based experiments and 2071–2100 for all ssp585-based experiments. Note that the baseline historical refers to realization r1i1p1f1.

| Experiment and response | Definition |
|---:|:---|
| *cloud* sspRes | ssp585-cloud−hist-cloud |
| *cloud* sspRes-diff | (ssp585-cloud−hist-cloud)−(ssp585−historical) |
| *eddy* sspRes | ssp585-eddy−hist-eddy |
| *eddy* sspRes-diff | (ssp585-eddy−hist-eddy)−(ssp585−historical) |
| *iceSheet* sspRes | ssp585-iceSheet−hist-iceSheet |
| *iceSheet* sspRes-diff | (ssp585-iceSheet−hist-iceSheet)−(ssp585−historical) |
| *snow* sspRes | ssp585-snow−hist-snow |
| *snow* sspRes-diff | (ssp585-snow−hist-snow)−(ssp585−historical) |
| *ozone* sspRes | ssp585-ozone−hist-ozone |
| *ozone* sspRes-diff | (ssp585-ozone−hist-ozone)−(ssp585−historical) |
| *piAerOxid* sspRes | ssp585-piAerOxid−hist-piAerOxid |
| *piAerOxid* sspRes-diff | (ssp585-piAerOxid−hist-piAerOxid)−(ssp585−historical) |
| ScenarioMIP range/sspRes-diff | (ssp585−historical)−(ssp126−historical) |

ssp585, we consider results from ssp126 and the SSPs corresponding to increased radiative forcing of 4.5 W m$^{-2}$ (ssp245) and and 3.7 W m$^{-2}$ (ssp370) by the end of the 21st century (O'Neill et al., 2016) carried out with NorESM2-MM for CMIP6 to

provide an uncertainty range associated with future scenarios. In addition, we have carried out experiments in which human influence on aerosols and aerosol precursors are excluded (hist-piAerOxid and ssp585-piAerOxid). For the historical period (hist-piAerOxid), we follow the protocol for the CMIP6 Aerosol Chemistry Model Intercomparison Project (AerChemMIP; Collins et al., 2017) experiment in which aerosols and aerosol precursors are set to pre-industrial levels and other forcings are as in the historical experiment. For the future period (ssp585-piAerOxid), we have designed an experiment corresponding to

hist-piAerOxid, but where other forcing agents evolve as in ssp585. The latter experiment is inspired by AerChemMIP, but is not a part of that protocol.

## 4 Results

In what follows, we consider how the different model updates affect future Arctic surface temperatures and investigate the processes driving these changes. We define the Arctic as latitudes poleward of 66°N.

While we evaluate the temporal evolution of the historical and future changes (1850–2100) for selected fields, the focus is primarily the *future response* (referred to as "sspRes" for short), computed as the difference between the last 30-years of the ssp585-based and historical-based experiments (2071–2100 and 1985–2014, respectively). The length of the time slices is in





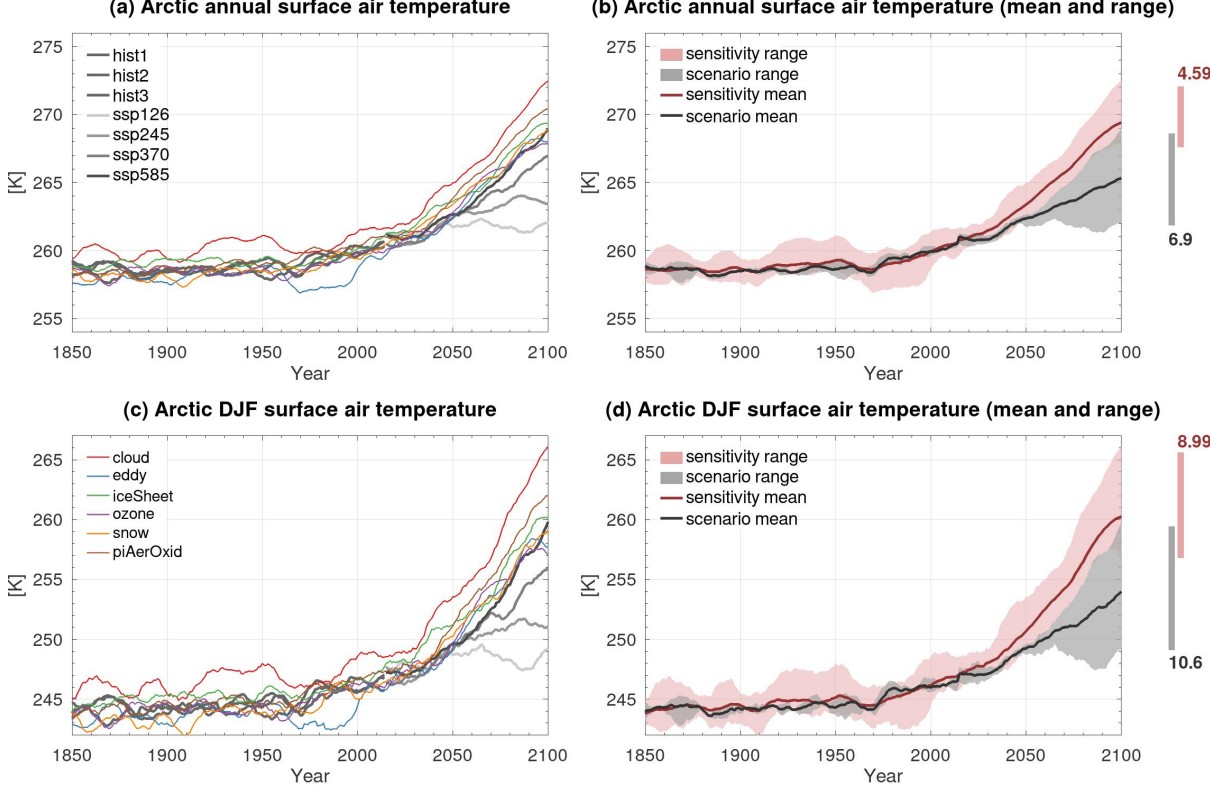

**Figure 1.** Time evolution of annual (a–b) and winter (c–d) Arctic surface temperature for 1850–2100. In a and c, the individual lines show the CMIP6 historical realizations (hist1, hist2, and hist3), four future scenarios (ssp126, ssp245, ssp370, ssp585), the NEEMS experiments (*cloud*, *eddy*, *iceSheet*, *ozone*, and *snow*), and *piAerOxid*. In b and d, the lines show the mean warming for the scenarios (grey) and sensitivity experiments (red) and the shading shows the warming range between the experiments with strongest and weakest warming. To highlight the end-of-the-century warming range, the difference between the experiment with strongest and weakest warming in 2100 is additionally shown in terms of the numbers and vertical bars to the right of b and d. A 10-year running mean has been applied. The legend for the lines in a and c is split between the two panels, however, both panels show all lines. Units are in K.

line with the World Meteorological Organization definition of the climatological standard normals. For the remainder of this paper, results are based on these time periods unless otherwise stated.

To highlight the difference between the sensitivity and baseline experiments, we also examine the *future response difference* ("sspRes-diff" for short), defined as the difference between two future response fields. A detailed overview of how the future responses and future response differences are defined is provided in Table 2.

## 4.1    Surface temperature

By 2100, NorESM2-MM projects annual-mean Arctic surface warming ranging from 2.76 to 9.66 K under ssp126 and ssp585,

relative to the pre-industrial state (Fig. 1a). Results for the individual seasons are shown in Fig. 1c and d for winter (December,





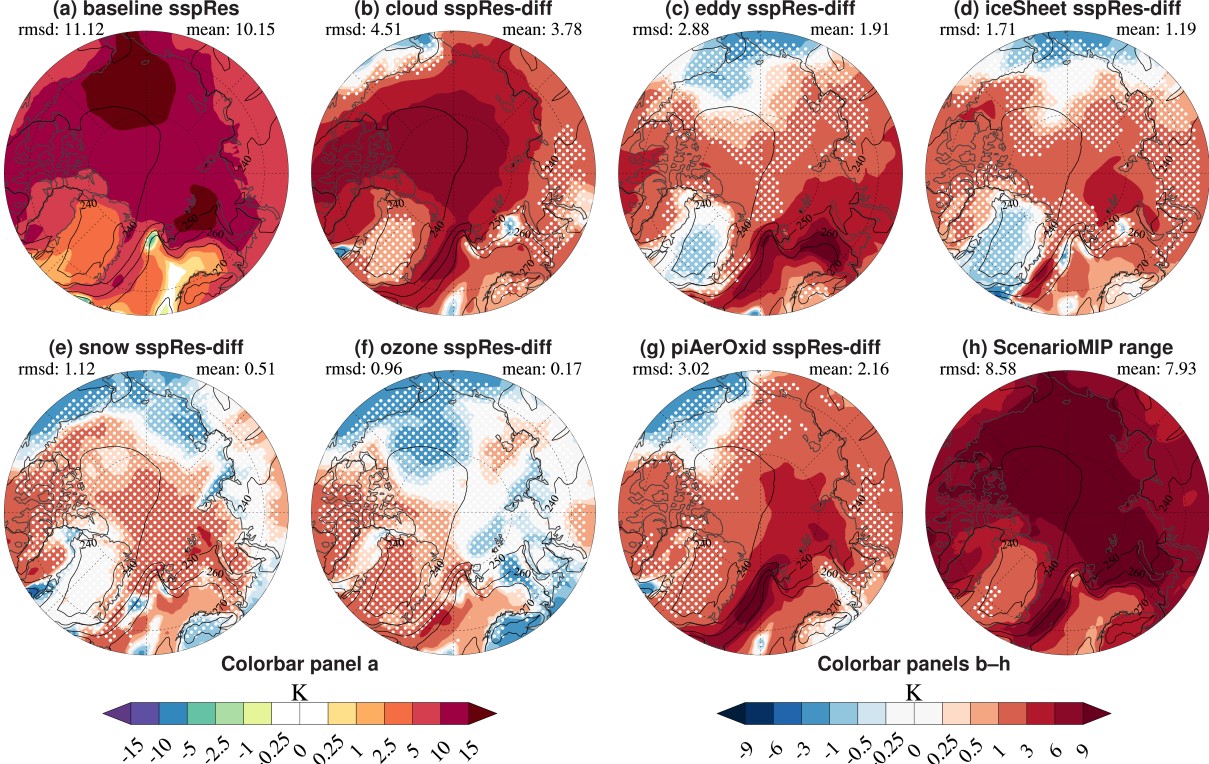

**Figure 2.** Spatial distribution of the future change in Arctic surface temperature (colors) during winter for the future response (sspRes; Table 2) for the baseline experiments (a) and the future response difference (sspRes-diff; Table 2) for *cloud* (b), *eddy* (c), *iceSheet* (d), *snow* (e), *ozone* (f), *piAerOxid* (g), and the ScenarioMIP range (h). The area average (mean) and the root-mean-square-difference (rmsd) of the field shown (colors) is given in the upper right and left corners of each panel. Note that the 30-year based mean value for the ScenarioMIP warming range shown here is slightly smaller than the year 2100 warming range in Fig. 1d (10.6 K) in line with temperatures generally increasing over the 21st century, rendering 2100 the warmest year on average. Selected contours from the historical baseline climatology (solid black lines) are shown for reference. White dots indicate *non-significant changes* according to a two-sided T-test. Note that we use different color scales and contour levels for a (left color bar) and b–h (right color bar), and that the contour levels are non-linear. Units are in K.

January, and February; DJF), and Fig. S1a–b for spring (March, April, and May; MAM), c–d for summer (June, July, and August; JJA), and e–f for autumn (September, October, and November; SON). The largest warming is projected during winter, ranging from 5.36 K in ssp126 to 15.94 K in ssp585 (Fig. 1c). In addition to having the largest warming, the winter season also exhibits the largest warming range (estimated as the difference between the warming in the most and least extreme scenarios considered here, ssp585 and ssp126, at year 2100), with a value of 10.6 K (Fig. 1d), which is nearly three times as large as the projected range for the summer warming (Fig. S1d; 3.39 K).





The sensitivity experiments consistently warm more than the baseline experiments, as seen by comparing the colored and black lines in Fig. 1a and c. There is moreover a notable difference between the sensitivity experiments that undergo the largest and smallest warming; by 2100 this difference is 4.59 K for the annual mean and 8.99 K for winter (red numbers and vertical bars to the right of Fig. 1b and d), which is almost as large as the ScenarioMIP warming range (6.9 K and 10.6 K; black numbers and vertical bars). The influence of the model updates is not only evident for the future projections, but can also be seen in the pre-industrial climate (as represented by years 1850–1900 in Fig. 1 and S1), where the sensitivity experiments have more pronounced spread than the three historical members (compare red and grey shading).

While the sensitivity experiments all exhibit additional Arctic warming compared to the CMIP6 baseline (Fig. 1), the details of the warming patterns vary from experiment to experiment (Fig. 2). The baseline warming (a) is characterized by surface temperatures exceeding 10 K over most of the Arctic and even 15 K over the Chukchi Sea. The *additional warming* (that is, the future response difference sspRes-diff; Table 2) seen in the sensitivity experiments (b–f) is strongest in the *cloud* experiments, which warm by 3.78 K on average for the whole Arctic region and by more than exceeding 6 K over the central Arctic Ocean and parts of Canada. The second and third largest additional warming is found in the *eddy* and *iceSheet* experiments, which warm by 1.91 K (c) and 1.19 K (d) on average with the largest warming found over the Nordic, Barents, and Kara Seas and over parts of Russia. The additional warming in *eddy* is similar in magnitude to the additional warming induced by excluding anthropogenic aerosols (2.16 K; g). The *ozone* and *snow* experiments exhibit the weakest additional warming (0.51 K and 0.17 K; e and f), with the regional changes mostly being non-significant (as indicated by the white dots).

## 4.2 Ocean temperature

Arctic warming is not only happening at the surface, but also within the Arctic Ocean, particularly in the upper 2000 m (e.g, Shu et al., 2022), with CMIP6 models indicating rapid and early detectable warming in the interior by the mid-21st century (Tjiputra et al., 2023). There also exists large model spread and divergence in the future Arctic Ocean warming as projected by CMIP6 models (Muilwijk et al., 2023).

In our NorESM2 simulations, the baseline winter warming (Fig. 3a and solid black line in Fig. 4) is strongest in the upper 1500 m, similar to in Shu et al. (2022) (their Fig. 1c), where the warm Atlantic Water is present, with the maximum warming occurring at 200–400 m depth. Moreover, the future warming is larger in the Eurasian Basin than in the Canadian Basin (compare Fig. 4a and b).

The sensitivity experiments are characterized by additional upper-ocean winter warming compared to the baseline (Fig. 3b–f). As with the baseline warming (Fig. 3a), the additional warming is strongest below the ocean surface, which indicates that the warming is likely associated with a warmer Atlantic Water inflow (which subducts below the sea surface in the Svalbard region). Similar to the atmosphere, the most pronounced surface warming is seen in the *cloud* experiment (b), with weaker warming in the *snow* (e) and *ozone* (f) experiments. The subsurface ocean in the *iceSheet* experiment (d and green lines in Fig. 4) warms to a similar extent as *cloud*, but the warming reaches greater depths south of 70°N. The vertical profiles for the historical and ssp585-based climatologies (Fig. S2) reveal that while the future *iceSheet* experiment is not among the warmest (compare dashed lines), the historical *iceSheet* experiment features a weaker Atlantic Water signature (that is, colder) than

Author(s) 2025




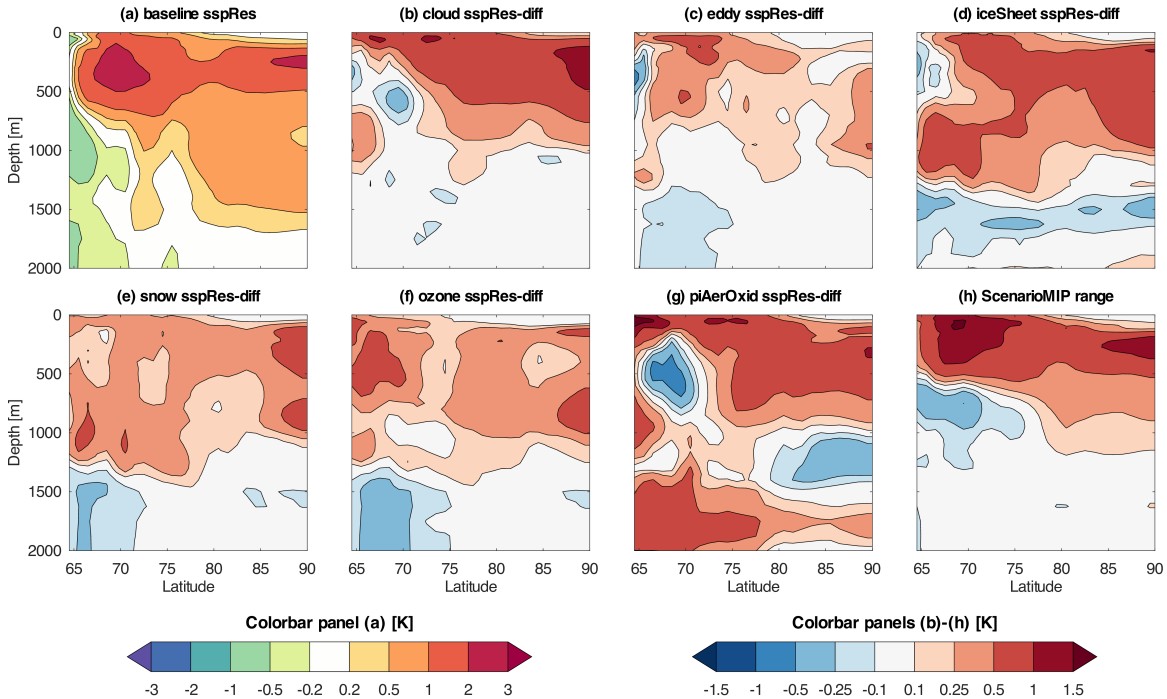

**Figure 3.** Future changes in zonal-mean Arctic ocean temperature during winter for the future response (sspRes; Table 2) for the baseline experiments (a) and the future response difference (sspRes-diff; Table 2) for *cloud* (b), *eddy* (c), *iceSheet* (d), *snow* (e), *ozone* (f), *piAerOxid* (g), and the ScenarioMIP range (h). Units are in K.

the other experiments, particularly in the upper 1000 m (solid lines), resulting in a large future response for *iceSheet* (Fig. 3 and 4). Understanding why the *iceSheet* experiment has a weak Atlantic Water signature in the present-day period is beyond the scope of the present paper, but we note that it does not appear to be linked to a larger weakening of the AMOC in this experiment compared to the others (Fig. S3). The additional ocean warming in the *eddy* experiment is relatively weak (Fig. 3c and purple lines in Fig. 4). The historical *eddy* experiment is very similar to the baseline, albeit slightly colder below 1000 m in the Eurasian Basin (Fig. S2), and the future *eddy* experiment undergoes very little additional future warming compared to the baseline experiment (Fig. S2). This differs from what we saw for surface temperature (Fig. 2), where *eddy* is considerably warmer than the baseline with a future response differences of 1.91 K.

One caveat is that in nearly all the historical runs, the Atlantic water temperature is cold-biased by about 1 K in both the Eurasian and the Canadian Basins (compare thin solid colored lines to thick black solid line in Fig. S2). This is a known issue with NorESM2 (e.g., Shu et al., 2023; Heuzé et al., 2023). The cold bias is similar across the baseline and the sensitivity runs, except for the historical *iceSheet* experiment (solid green lines) which is even colder, and the historical *piAerOxid* experiment (solid brown lines) which has a thicker layer of the Atlantic water (and equally cold-biased).





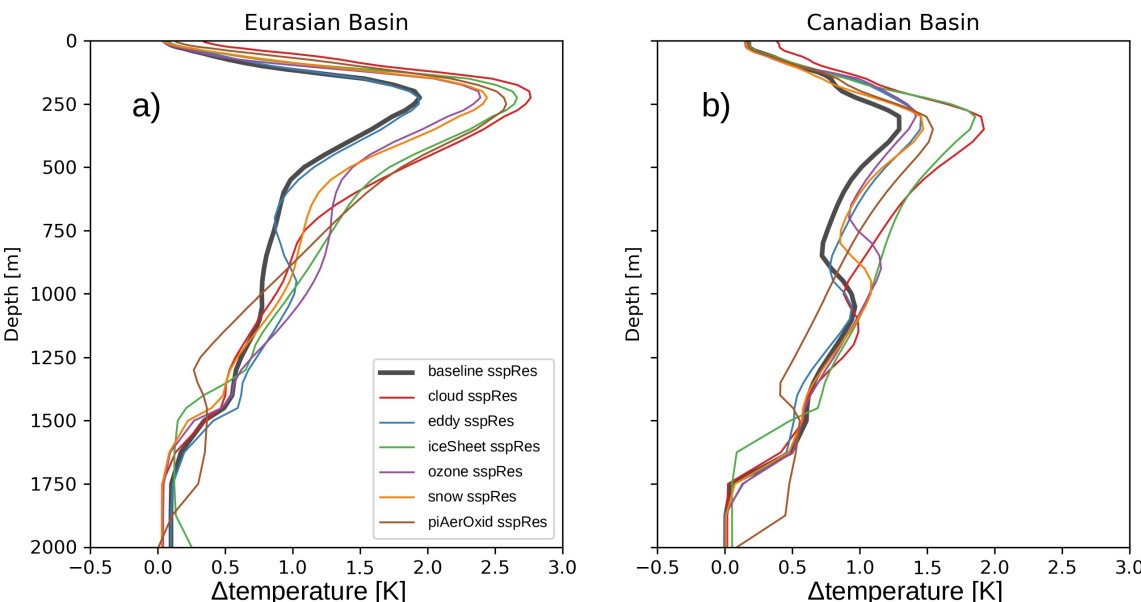

**Figure 4.** Future changes in vertical ocean temperature profiles averaged for the Eurasian (a) and Canadian Basins (b) during winter. Both panels show the future response (sspRes; Table 2) for the baseline (grey line), *cloud* (red line), *eddy* (blue line), *iceSheet* (green line), *ozone* (purple line), *snow* (orange line) and *piAerOxid* (brown line). Units are in K.

### 4.3 Sea ice

Arctic sea-ice loss is a key driver of Arctic warming. Examining the future changes in Arctic winter sea-ice volume in the baseline simulations and the future response anomalies in the sensitivity experiments reveals that the largest additional reduction in sea-ice volume is found in the *eddy* experiment, followed by the *ozone* and *iceSheet* experiments (Fig. 5c, e, and d respectively). Somewhat surprisingly, the *cloud* experiments, which undergo the strongest surface warming (Fig. 2b), display the weakest changes in sea-ice volume: a reduction over parts of the central Arctic Ocean, Laptev Sea, and East Siberian Sea (Fig. 5b).

The largely weak future response anomalies in *cloud* indicate that the change in sea-ice volume between the present-day and future periods are very similar to that in the baseline experiments. The *snow* and *piAerOxid* experiments (f and g) stand out from the rest with a smaller future reduction (that is, a positive future response difference) in sea-ice volume than the baseline experiments (a).

    To understand why the sea ice is changing the way it is in the different sensitivity runs, it is crucial to consider these changes

both in the context of how the sea-ice regime shifts in the future period, and in the context of the realism of the respective present-day state. It is well-known that the historical baseline experiment is characterized by the Northern Hemisphere (NH) sea ice being too thick, which in turn results in the reduction in summer sea ice being too slow (Seland et al., 2020b). Having a realistic representation of sea ice compared to observations is a necessary prerequisite for being able to capture important aspects of both the present-day Arctic climate and the future changes.



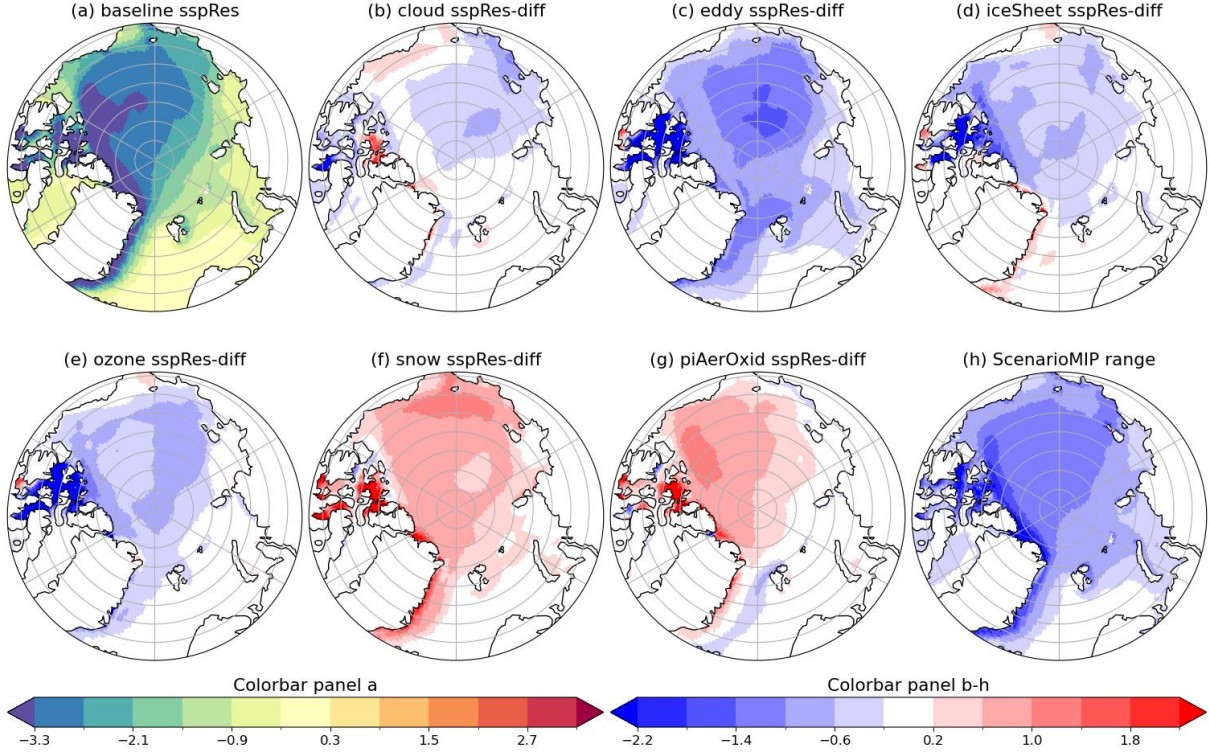

**Figure 5.** Spatial distribution of future changes in sea-ice volume during winter for the future response (sspRes; Table 2) for the baseline experiments (a) and the future response difference (sspRes-diff; Table 2) for *cloud* (b), *eddy* (c), *iceSheet* (d), *ozone* (e), *snow* (f), *piAerOxid* (g), and the ScenarioMIP range (h). Units are in meter per unit area.

To assess the performance of the sea-ice area and volume in the different experiments, we consider the area in September, when it is typically at its lowest, and the volume in March, when it is typically at its largest, in line with the community standards. The time evolution of September sea-ice area (Fig. 6a) and March sea-ice volume (b) shows that all of the ssp585-simulations (baseline and NEEMS) are practically ice free (ice area below 1 mill km$^2$) in September in what we have defined as the future response period (2071–2100, see Sect. 4). Thus, the future sea-ice volume in these simulations only reflects growth

since last fall, and the variation in volume between the simulations are then much smaller than earlier in the simulations.

    Figure 6 also reveals that modifications of snow conductivity applied in the *snow* experiments (Sect. 3.4) successfully brings NH sea-ice area and volume more in line with observational estimates from the Ocean and Sea Ice Satellite Application Facilities (OSI SAF) reprocessed data (EUMETSAT, 2022b, a) and both Pan-Arctic Ice-Ocean Modeling and Assimilation System (PIOMAS) model-based time series (Schweiger et al., 2011) and satellite-based estimates from CryoSat-2 (Tilling

et al., 2018). In contrast, all other sensitivity experiments have thicker sea ice during the period with observations, and in general, even thicker sea ice in the pre-industrial climate. Of the other sensitivity experiments, *piAerOxid* and *cloud* are the third and second closest to the volume observations respectively, and these are both characterized by a larger warming in the





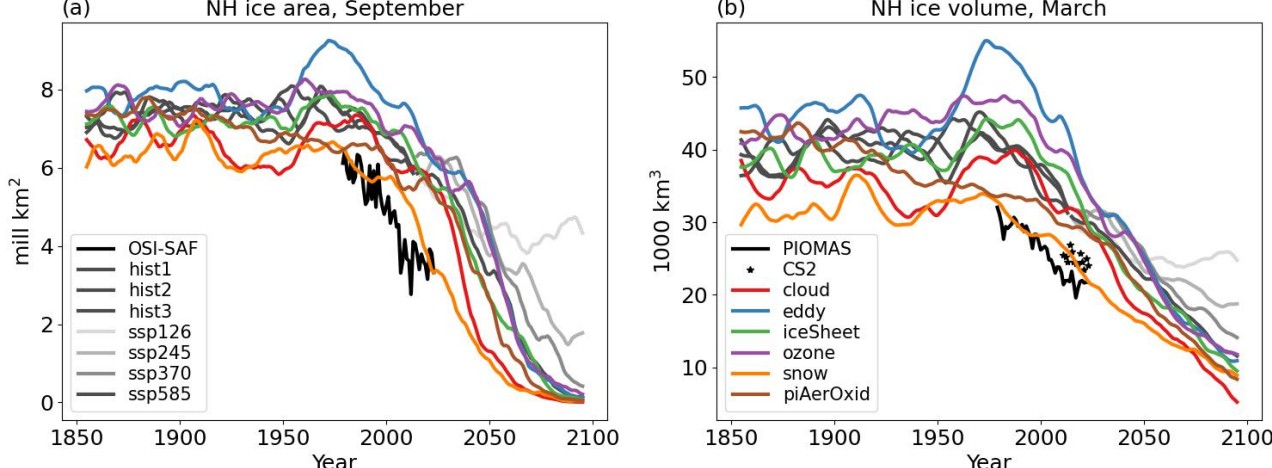

**Figure 6.** Time evolution of NH sea-ice area in September (when the area is typically at its lowest; a), and sea-ice volume in March (when the volume is typically at its largest; b) from CMIP6 historical (three realizations; hist1, hist2, and hist3), four future scenarios (ssp126, ssp245, ssp370, and ssp585), and the five NEEMS simulations (*cloud*, *eddy*, *iceSheet*, *ozone*, and *snow*), and *piAerOxid* for 1850–2100. Also shown in black are the OSI-SAF satellite record of sea ice area (black line in a), and the sea-ice volume estimates by PIOMAS (black line in b), and satellite estimates from CryoSat-2 (black stars in b). Annual values are shown for the observations opposed to the 10-year running mean used for the model simulations, for clarity. Summer sea-ice area (September) is shown to illustrate the transition to and seasonal Arctic ice cover, while March sea-ice volume represents the seasonal sea-ice maximum. Satellite-based ice-volume estimates are not available during summer. Units are in mill km$^2$.

historical period than the other experiments. This can also be seen in the spatial distribution of present-day sea-ice volume, which shows too much ice in the marginal ice zones and a thickness bias in the Beaufort Sea (Fig. S4).

The *eddy* experiment, which has the strongest *additional* reduction in sea-ice volume of the sensitivity experiments (Fig. 5c), has a pronounced maximum in integrated Arctic sea-ice volume during the last half of the 20th century (Fig. 6b), which exceeds the maximum volume of all other experiments shown for the whole time period (1850–2100). Similar behavior is seen for the other sensitivity experiments that are characterized by negative future response differences (that is, even stronger reductions in sea-ice volume than the baseline), the *ozone*, *cloud*, and *iceSheet* experiments (Fig. 5e, b, and d), although less pronounced.

The simulations with the smallest sea-ice volume in the historical period (*snow* and *piAerOxid*) have less ice than the CMIP6 baseline (Fig. S4b, g, and h and Fig. 6b), which explains why the future response difference is positive for these experiments (Fig. 5f and g). Due to the lack of perennial ice at the end of the 21st century, and consequently very little variation between the simulations, the loss of sea-ice volume is a strict function of the amount of sea ice in the reference (historical) period (1985–2014). When sea ice is perturbed during the growth season, it returns to its equilibrium thickness by adjusting growth rate. This

is know as the growth-thickness feedback, which is much stronger for thinner than thicker ice (Bitz and Roe, 2004). The large variability in sea-ice volume in the historical period, both inter-annually and between the experiments, can be understood by



this negative feedback. Perturbations and variations are amplified in the historic period relative to the future period where the ice is much thinner.

Although the *snow* simulation, which is closest to the observations in terms of both sea-ice area and volume, reaches almost
ice-free conditions in September before all other experiments, the *cloud* experiment, which is much warmer, exhibits faster sea-ice loss and ends up having a lower September sea-ice area and March sea-ice volume in the later decades of the 21st century than any other experiment (Fig. 6). The *cloud* experiment also shows a larger reduction in sea-ice area during winter (Fig. S5), where larger reductions are associated with larger additional surface warming (Sect. 4.1). The *snow* experiment modifies the negative growth-thickness feedback by reducing the amount of thicker sea ice, but also at the same time, very
effectively isolating the atmosphere from the warm ocean. The cold atmosphere above the ice promotes a rapid refreezing of leads and openings during winter.

### 4.4   Spatio-temporal characteristics of surface warming

The future surface warming in the sensitivity experiments exhibits variations in space (Fig. 2) and time (Fig. 1). To explore the dominant patterns and characteristics of these variations, we use Empirical Orthogonal Function (EOF) analysis. The leading
EOFs of surface temperature and other fields were identified in the baseline experiment using the entire period 1850–2300 (historical + ssp585 + extension to 2300). The corresponding modes of variability in the sensitivity experiments were obtained by projecting anomalies of the same fields, taken over the period 2000–2100, onto the baseline EOFs. This analysis was performed both for the Arctic, north of 66°N, and for the NH extratropics, north of 20°N. The fields analysed are surface temperature and the surface heat flux (the sum of the surface latent and sensible heat fluxes). The EOF analysis was carried out
on winter means, with a 10-year low-pass filter and spatial smoothing with a scale of approximately 500 km.

We focus on the first and second modes of variability (EOF1 and EOF2), since these are strongly associated with scenario-forced warming towards equilibration, and regionally accelerated or decelerated warming which is more transient (Keil et al., 2020; He et al., 2022), respectively. For the baseline experiment, EOF1 explains 98.4% and 98.9% of the variance in the surface warming over the NH and the Arctic (Fig. 7a and c), with Arctic amplification in a spatial pattern of uniform sign associated
with areas of highest climatological sea-ice concentration. The associated baseline time series indicates that EOF1 represents Arctic warming that accelerates at the turn of the 21st century and begins to equilibrate around 2200 (Fig. 8a, black line). EOF2 explains only 0.4% of the variance over the Arctic and NH, with opposite-signed centres of action at high latitudes over the Atlantic sector and the Kara and Chukchi seas (Fig. 7b and d). The signal in the Pacific sector reflects changes in atmospheric circulation there, in particular a transient relative strengthening of the Aleutian low in EOF2 of the sea level pressure field (not
shown). The time series associated with EOF2 represents an accelerated warming around the sea-ice edges of the Kara and Chukchu seas that peaks around 2100 (Fig. 8b, black line).

Differences between experiments are more easily seen from Fig. 9, which shows the trends over the time period 2000–2100 and their significance (in terms of the signal-to-noise ratio) for the different EOFs. The sensitivity experiments generally exhibit enhanced warming towards equilibration (EOF1) over the period 2000–2100 compared to the baseline experiment (red
cells in Fig. 9), while their transient warming pattern (EOF2) may be amplified or suppressed according to the experiment





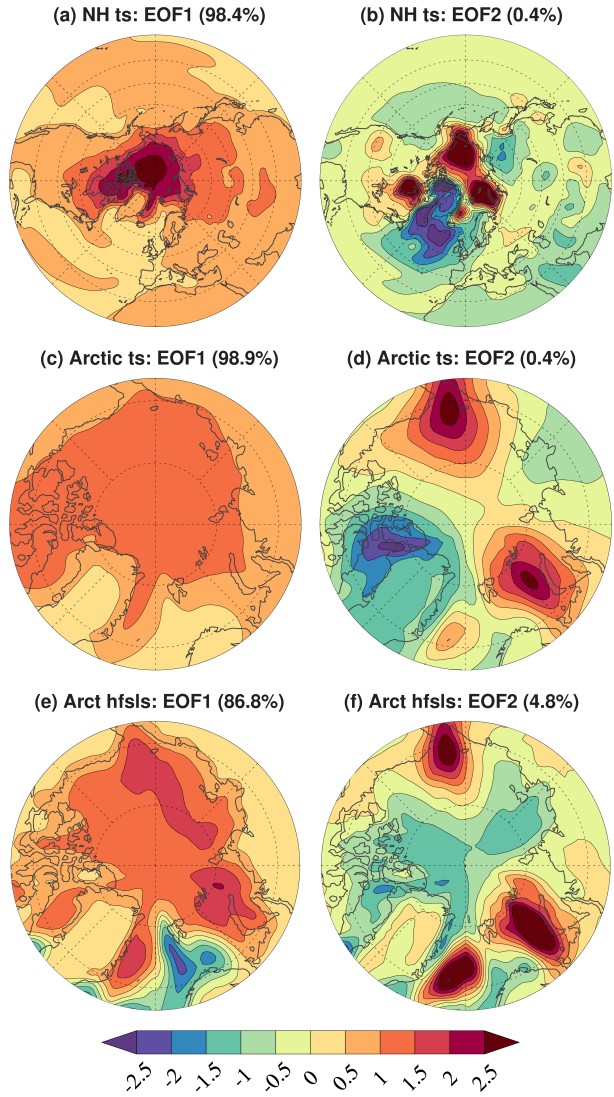

**Figure 7.** Spatial distribution of the first and second EOFs (EOF1 and EOF2) based on NH surface temperature (ts; a and d), Arctic surface temperature (ts; c and e) and Arctic total surface heat fluxes (hfsls; c and f) for the baseline experiments. The EOFs are shown for the region for which they are computed. Note b–f show the Arctic region whereas a–b show the NH extratropics. The variance explained associated with each EOF is given in parenthesis in the figure titles. The fields share the same color bar; all fields are normalized by their standard deviation and thus unitless.

(red or blue cells). The most rapid Arctic warming towards equilibration (EOF1) is found in the *cloud* experiment, suggesting that it would saturate at a warmer equilibrium temperature than the baseline. The only experiment exhibiting slower warming than the baseline for EOF1 is *piAerOxid*. Arctic EOF2-related warming is stronger than the baseline in some experiments (*eddy*, *iceSheet*, *ozone*), but weaker in others (*cloud*, *snow*), suggesting the latter exhibit a less prominent lag in North Atlantic





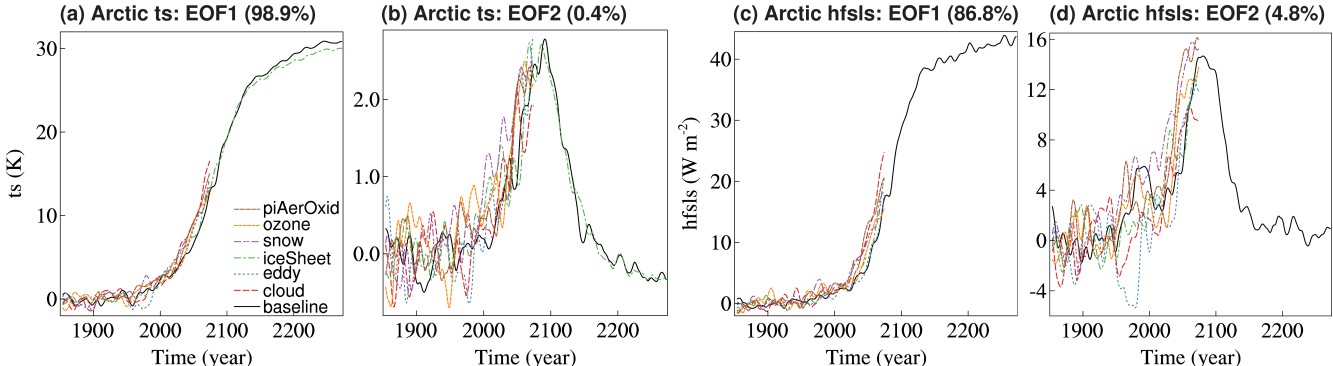

**Figure 8.** Time series associated with the first two EOFs for Arctic surface temperature (ts; a–b) and surface heat fluxes (hfsls; c–d) for all experiments (baseline, NEEMS, and *piAerOxid*). A 10-year running mean has been applied. The variance explained associated with each EOF is given in parenthesis in the figure titles. All fields are unitless.

|  |  | Ref | cloud | eddy | iceSh | snow | ozone | piAer |
|---|---|---|---|---|---|---|---|---|
| ts NH | EOF1 98.4% | 0.06 | 0.08 | 0.08 | 0.07 | 0.07 | 0.07 | 0.02 |
|  | EOF2 0.4% | 0.01 | 0.01 | 0.01 | 0.01 | 0.01 | 0.01 | -0.01 |
| ts Arct | EOF1 98.9% | 0.12 | 0.16 | 0.14 | 0.13 | 0.13 | 0.15 | 0.03 |
|  | EOF2 0.4% | 0.03 | 0.02 | 0.03 | 0.03 | 0.02 | 0.03 | -0.00 |
| hfsls Arct | EOF1 86.8% | 0.15 | 0.24 | 0.16 | 0.18 | 0.15 | 0.21 | 0.17 |
|  | EOF2 4.8% | 0.09 | 0.12 | 0.17 | 0.13 | 0.13 | 0.17 | 0.12 |

**Figure 9.** Overview of EOF trends (numbers) and their significance (colors) for surface temperature (ts) for the NH extratropics and Arctic (Arct) and for the surface heat flux (hfsls) for the Arctic. The fields and regions are given in column 1, the EOFs and their associated variance explained are shown in the second column, and the EOF trends and their significance for the baseline, *cloud*, *eddy*, *iceSheet* (iceSh), *snow*, *ozone*, and *piAerOxid* (piAer) experiments are shown in columns 3–9. For all experiments, the relevant historical and scenario runs are combined to provide a long time period from 1850 to 2100. For each experiment and EOF, the trend is computed for years 2000–2100 and the associated signal-to-noise ratio (SNR), defined as the ratio between the trend and the variance, for the historical period (1900–2000). Trends are considered significant when the SNR values are 1 or larger. For columns 3–9, white cells indicate insignificant trend values. For column 3, yellow cells indicate significant baseline trends. For columns 4–9, blue cells indicate that the trends are significant and smaller than the baseline trend, while red cells indicate that the trends are significant and larger than the baseline trend.



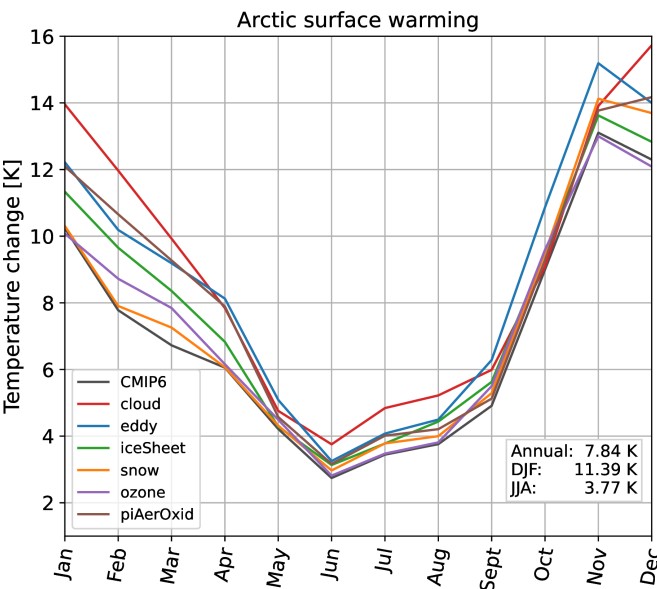

**Figure 10.** Monthly evolution of the future response (sspRes; Table 2) in Arctic surface temperature for the CMIP6 baseline (black line), *cloud* (red line), *eddy* (blue line), *iceSheet* (green line), *snow* (orange line), *ozone* (purple line), and *piAerOxid* (brown line). The annual, DJF, and JJA mean values, averaged across the baseline and five NEEMS experiments (*cloud*, *eddy*, *iceSheet*, *snow*, and *ozone*), are shown in the right legend. Units are in K.

warming (described as an "Atlantic warming hole", c.f. Keil et al., 2020; He et al., 2022). It is possible that the weaker warming in *cloud* and *snow* is due to the longer equilibration time scales in these experiments, but this cannot be assessed due to the duration of the experiments (1850–2100).

Changes in the surface heat fluxes closely track the changes in Arctic warming, both in terms of the spatial pattern (Fig. 7e and f) and the temporal evolution (Fig. 8c and d). In terms of the trends, they exhibit a significant increase in EOF1 and EOF2 for all experiments compared to the control. In the next section, we further investigate how surface heat fluxes and other key processes contribute to Arctic surface warming. We study the mean behavior across the NEEMS and baseline experiments, establish the regions where the range in responses across the experiments is large, and identify the processes associated with the spread.

### 4.5 Surface temperature decomposition

The total future Arctic surface warming is characterized by pronounced seasonal variations (Fig. 10), with the smallest warming occurring in June and largest in November–December. The spread between the sensitivity experiments (colored lines) is small during summer and fall, particularly between September and mid-November, and increases substantially during the cold months. All the modified experiments undergo warming that is comparable to or larger than the baseline warming (black line) throughout the year, and especially in DJF.





To identify the processes dominating the additional Arctic warming seen in the sensitivity experiments, we linearly decompose the total surface warming by analyzing changes in the surface energy fluxes, following Lu and Cai (2009) and Boeke and Taylor (2018). The surface energy budget can be computed as

$$Q = (1 - \alpha)\text{SW} \downarrow + \text{LW} \downarrow - \epsilon\sigma T_s^4 - (\text{SH} + \text{LH}), \tag{1}$$

where $Q$ represents the storage of heat across all surface types, as well as oceanic heat transport, SW↓ is the downwelling shortwave flux, $\alpha$ is the surface albedo defined as the ratio of upward to downward shortwave clear-sky fluxes, $\sigma$ is the Stefan-Boltzmann constant, $T_s$ is the surface temperature, LW↓ is the longwave downwelling radiation, $\epsilon\sigma T_s^4$ is the upwelling longwave radiation emitted from the surface (emissivity $\epsilon$ is assumed to be 1), and LH and SH are the surface fluxes of latent and sensible heat. Solving for $T_s$, the total change in surface temperature can be approximated by:

$$
\begin{aligned}
\Delta T_s = (4\sigma T_s^3)^{-1}\Big( & -\Delta\alpha(\text{SW} \downarrow + \Delta\text{SW} \downarrow) \\
& + (1 - \Delta\alpha)(\text{SW} \downarrow_{\text{CLD}} + \Delta\text{LW} \downarrow_{\text{CLD}} \\
& + \Delta\text{SW} \downarrow_{\text{CLR}}) + \Delta\text{LW} \downarrow_{\text{CLR}} - \Delta Q \\
& - \Delta(\text{SH} + \text{LH})\Big).
\end{aligned} \tag{2}
$$

Hence, the total temperature change can be approximated by the sum of the temperature contributions associated with six different processes:

1. changes in surface albedo feedback (SAF)

$$\frac{-(\Delta\alpha)(\text{SW} \downarrow + \Delta\text{SW} \downarrow)}{4\sigma T_s^3}, \tag{3}$$

2. changes in cloud radiative effect (CRE)

$$\frac{(1 - \Delta\alpha)(\text{SW} \downarrow_{\text{CLD}} + \Delta\text{LW} \downarrow_{\text{CLD}})}{4\sigma T_s^3}, \tag{4}$$

3. changes in non-SAF shortwave clear-sky radiation (SWCS)

$$\frac{(1 - \Delta\alpha)\text{SW} \downarrow_{\text{CLR}}}{4\sigma T_s^3}, \tag{5}$$

4. changes in longwave clear-sky radiation (LWCS)

$$\frac{\Delta\text{LW} \downarrow_{\text{CLR}}}{4\sigma T_s^3}, \tag{6}$$

5. changes in ocean heat storage and transport (HSTORE)

$$\frac{-\Delta Q}{4\sigma T_s^3}, \tag{7}$$





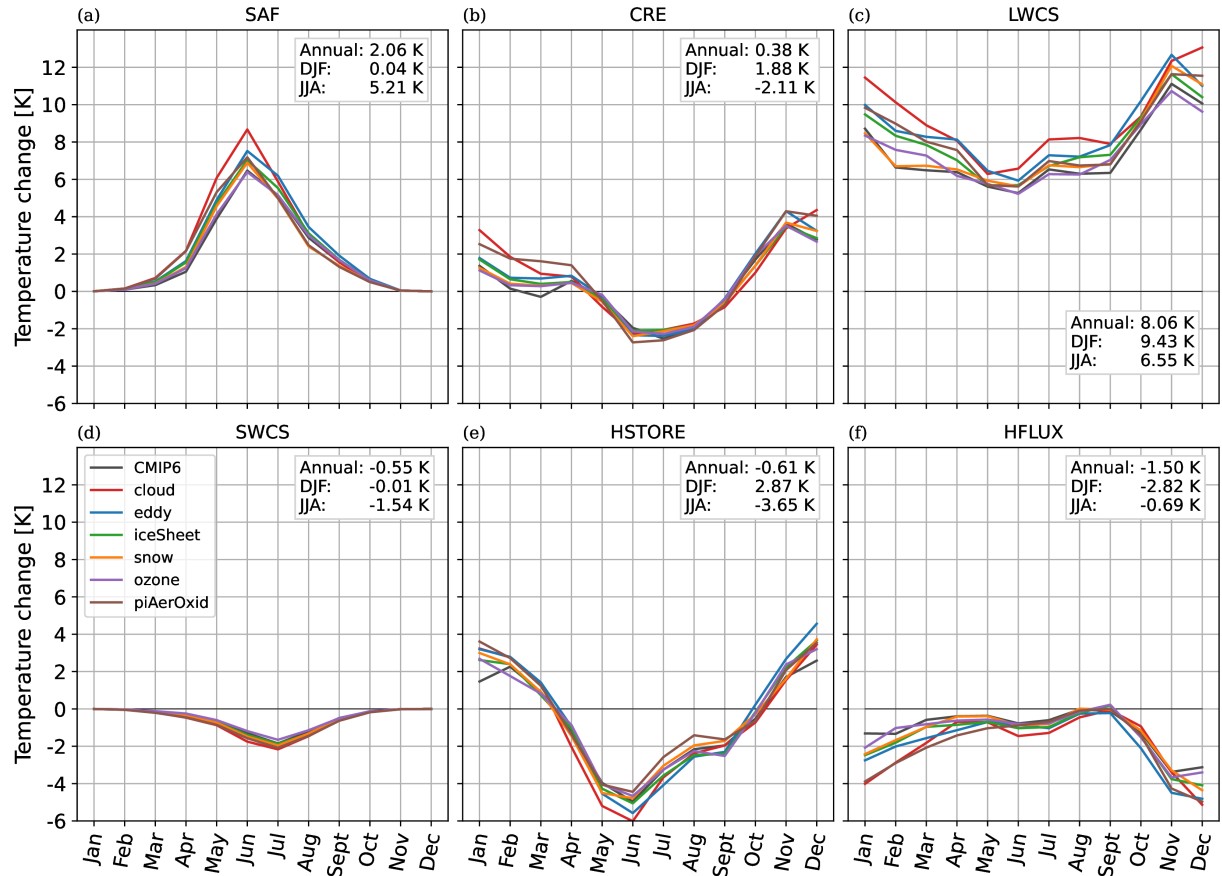

**Figure 11.** As in Fig. 10, but for the evolution of the future response (sspRes; Table 2) in the surface-warming decomposition terms SAF (a), CRE (b), LWCS (c), SWCS (d), HSTORE (e) and HFLUX (f). Units are in K.

6. and changes in latent and sensible heat fluxes (HFLUX)

$$\frac{-\Delta(\mathrm{SH}+\mathrm{LH})}{4\sigma T_s^3},\tag{8}$$

where $\mathrm{SW}{\downarrow}$, $\mathrm{SW}{\downarrow}_{\mathrm{CLR}}$, and $\mathrm{SW}{\downarrow}_{\mathrm{CLD}}$ are the downwelling shortwave fluxes for full-sky, clear-sky, and cloudy conditions

(with the latter defined as the difference between the full-sky and clear-sky fluxes) and $\mathrm{LW}{\downarrow}$, $\mathrm{LW}{\downarrow}_{\mathrm{CLR}}$, and $\mathrm{LW}{\downarrow}_{\mathrm{CLD}}$ are the longwave counterparts. The LWCS contribution accounts for the effects of air temperature and water-vapor changes from both local and remote sources, the HSTORE term represents changes in ocean heat storage and transport and is calculated as a residual, and HFLUX, the turbulent fluxes, is considered positive from the atmosphere to the ocean. The units of the six temperature-contribution terms (SAF, CRE, SWCF, LWCS, HSTORE, and HFLUX) is K.

The temperature decomposition reveals that marked seasonal variability evident in Fig. 10, is also seen for several processes (Fig. 11), in particular SAF (a), CRE (b) and HSTORE (e). This means that different seasons are to some extent characterized by different processes. The Arctic warming is dominated by LWCS (c) which is positive throughout the year and largest during





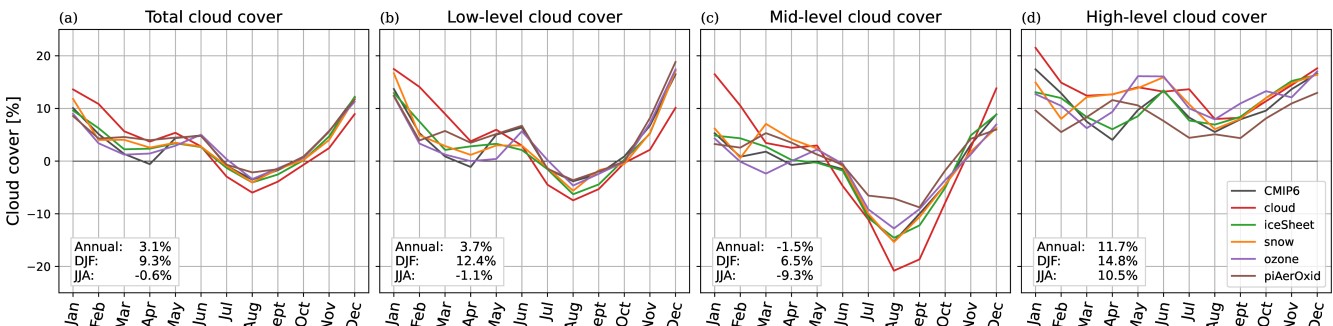

**Figure 12.** As in Fig. 10, but for the monthly evolution of the Arctic future response (Table 2) in total cloud cover (a), low-level cloud cover (b), medium-level cloud cover (c), and high-level cloud cover (d). Units are in percentage change.

late fall and early winter. During summer, the warming is predominantly caused by SAF and LWCS, and counteracted by CRE, SWCS (d), and HSTORE which act to cool the surface during the warm season. Arctic winter warming is mainly caused
by LWCS, with contributions from CRE and HSTORE. In what follows, we consider the main characteristics of the different terms in more detail.

### 4.5.1 Albedo

While the surface albedo feedback is negligible in the wintertime due to lack of sunlight, it has the second highest warming contribution during summer (legends in Fig. 11). Arctic warming due to SAF (Fig. 11a) peaks in June, driven by high insolation
and reduced sea ice and continental snow cover. This warming occurs throughout the Arctic but is especially pronounced in regions with significant sea-ice reduction, such as the Greenland-Barents Seas and the Beaufort-Chukchi Seas (Fig. S6a). The summer surface warming from decreased albedo, averaging at 5.21 K (legend in Fig. 11a), is partly offset by a combined cooling effect of $-3.65$ K from increased reflectivity by clouds (CRE; Fig. 11b) and aerosols (SWCS; Fig. 11d).

### 4.5.2 Clouds

The temperature contribution due to changes in cloud radiative effect (CRE) is associated with Arctic warming during winter, fall and early spring, and cooling during late spring and summer (Fig. 11b). Consistent changes are seen when considering the Arctic total cloud cover (Fig. 12a), which increases in all simulations throughout most of the year, except in August and September. The latter is due to changes in the low and mid-level clouds (Fig. 12b–c), with the strongest changes occurring over land (Fig. S7j–k). While this reduction is strongest for the mid-level clouds when considering the Arctic mean changes
(Fig. 12c), the low-level clouds undergo the strongest reductions regionally (Fig. S7j–k). However, over the central Arctic Ocean, the low-level cloud cover increases across all seasons (Fig. 12b and Fig. S7b, f, and j). The wintertime increase in cloudiness is consistent with a gradual transition from ice to liquid with warming, which is expected to prolong cloud lifetime




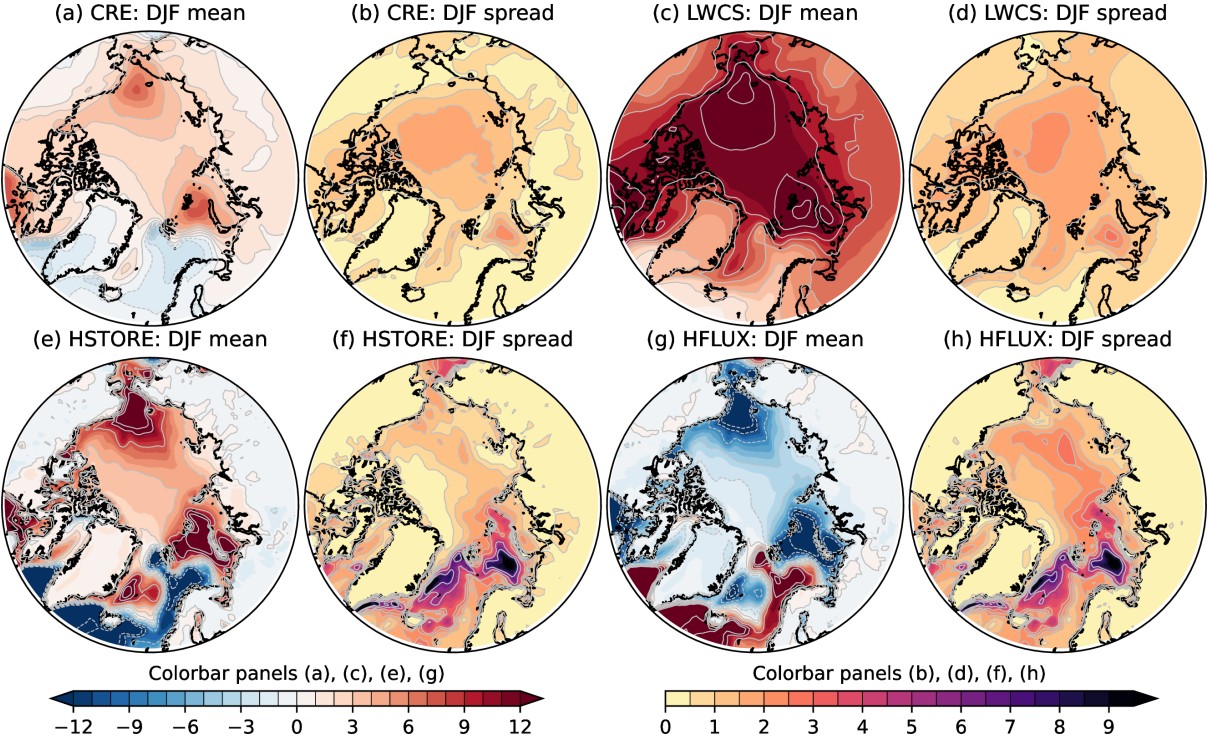

**Figure 13.** Spatial distribution of winter ensemble mean (a, c, e, and g) and spread (b, d, f, and h), computed across the baseline and the NEEMS experiments (*cloud*, *eddy*, *iceSheet*, *snow*, and *ozone*), for the future response (Table 2) in the surface temperature decomposition terms: CRE (a–b), LWCS (c–d), HSTORE (e–f), and HFLUX (g–h). Note SAF and SWCS are not shown as these terms are negligible during winter. Units are in K.

(e.g., Tan and Storelvmo, 2019). Reduced cloud cover over land has also been found in observations, and has been attributed to reduced near-surface relative humidity in the same regions (Liu et al., 2023).

During winter, changes in CRE enhance the greenhouse effect and surface warming (Fig. 11b and Fig. 13a). Regionally, the largest increase is found in the sea-ice retreat regions with maxima in the Barents-Kara and Beaufort-Chukchi Seas (Fig. 13a). This is due to changes in low-level clouds (Fig. 14k/Fig. S7b) which dominate the changes in the total cloud cover (Fig. S7a–d). While low-level clouds dominate the change pattern in total cloudiness during winter (Fig. S7), mid- and high-level clouds are also changing (Fig. 12c–d). The mean Arctic changes in high-level winter cloudiness are comparable to that occurring at low

levels (14.8% vs 12.4% increase; legends in d and b) whereas the mid-level changes are slightly smaller (6.5% increase; legend in c). The high-level cloud cover increases throughout the year with little seasonal variation and has a relatively uniform spatial pattern compared to the low-level cloud cover (Fig. 12d and Fig. S7d, h, and l).





**Figure 14.** As in Fig. 13, but for surface temperature (ts; a–b), latent heat fluxe (hfls; c–d), sea-ice concentration (siconc; e–f), sea-ice volume (sivol; g–h), near-surface specific humidity (huss; g–h), and low-level cloud cover (cldlow; k–l). Units are in K (a–b), W m$^{-2}$ (c–d), m (g–h), kg kg$^{-1}$ (g–h), and fraction (k–l).





### 4.5.3 Ocean heat storage and surface heat fluxes

The Arctic Ocean absorbs most of the excess summer heat, with the temperature contribution due to changes in ocean heat
storage (HSTORE; Fig. 11e) largely mirroring SAF (Fig. 11a) but at a lower magnitude. HSTORE is negative from April to
October across all runs, reflecting the accumulation of energy in the upper ocean. This warmer upper ocean delays sea-ice
formation in the fall and provides an energy source for the atmosphere during the Arctic winter. Overall, the Arctic-mean
surface warming from HSTORE is 2.87 K during winter and $-3.65$ K during summer. The annual surface cooling from
HSTORE is $-0.61$ K, thus representing a transfer of energy from the atmosphere to the ocean.

The surface cooling from turbulent fluxes (HFLUX; Fig. 11f) is small during the Arctic summer but becomes significant
(up to $-4$ K) in fall and winter with HFLUX mirroring HSTORE. The HFLUX changes represent a positive flux of energy
from the surface (i.e., mainly ocean) to the lower atmosphere which act to cool the surface and warm the overlying air. Hence,
resulting in a negative surface temperature contribution. HFLUX controls the total rate of surface cooling in the Arctic winter,
as all other terms contribute to warming or are negligible due to the lack of incoming sunlight (Fig. 11 and 13). The surface
temperature contribution from HFLUX is $-2.82$ K during winter (legend in Fig. 11f), thus more or less cancelling the warming
contribution from HSTORE (2.87 K; legend in e). This cancellation is also seen regionally, as the pattern of the changes for
HSTORE and HFLUX are very similar, only with opposite signs (Fig 13e and g). As seen for the total surface winter warming
(Fig. 14a), the HSTORE warming (Fig. 13e) is strongest in the sea-ice retreat regions (Fig. 14e). Conversely, the HFLUX
cooling (Fig. 13g) is also strong in the same regions. So while HSTORE (i.e., ocean heat storage and transport) contributes to
surface warming, HFLUX cools the surface and warms the lower atmosphere.

### 4.5.4 Longwave radiation

While both CRE and HSTORE contribute to Arctic surface warming during winter (1.88 K and 2.87 K; Fig. 11b and e), the
greatest contributor is the temperature change due to increased longwave downwelling radiation from the lower atmosphere
(LWCS), with an overall Arctic-mean winter warming of 9.43 K (Fig. 11c and Fig. 13c). While the strongest warming is found
in winter, LWCS contributes to warming throughout the year with an annual mean warming of 8.06 K, hence dominating the
Arctic-mean annual warming as well as the winter warming. Several mechanisms contribute to this warming. The warmer
ocean increases evaporation (Fig. 14c), which is also reflected in HFLUX during fall and winter (Fig. 13g). Near-surface
specific humidity increases across the Arctic (Fig. 14i), particularly in the sea-ice retreat regions (Fig. 14e), aligning with the
largest increase in winter low-level cloudiness (Fig. 14k). The increased humidity in turn contributes to enhanced downward
long wave radiation at the surface, reflected in the aforementioned surface contribution from LWCS (Fig. 11c, 13c).

### 4.5.5 Ensemble spread

Above, we considered the ensemble-mean changes in the surface-temperature decomposition terms, averaged across the base-
line and NEEMS experiments. There is however considerable spread in the Arctic-mean surface warming due to CRE, LWCS,
HSTORE, and HFLUX between the individual experiments, and this is particularly evident when considering the future re-




**Figure 15.** Overview of winter future response differences (sspRes-diff; Table 2) for the surface-temperature decomposition terms CRE, LWCS, HFLUX, and HSTORE. Panel a) shows changes in the four terms for *cloud* (red bars), *eddy* (blue bars), *iceSheet* (green bars), *snow* (orange bars), *ozone* (purple bars), *piAerOxid* (brown bars), and the ScenarioMIP range (black bars). For reference, the future response (sspRes; Table 2) for the baseline (grey values) is also shown. In panel b), the stacked bars show the surface-temperature contributions associated with CRE (light green), LWCS (light yellow), HSTORE (light purple), and HFLUX (light red) for same experiments as in a. Note that whereas all bars are annotated in a (numbers outside the outer edge of each bar), categories in b are only annotated (black numbers within bars) when their absolute values exceed 0.15 for readability. The terms SAF and SWCS are not shown as values are negligible. Units are in K.

sponse differences (Fig. 15). Changes are generally largest for the *cloud* experiment (red bars in Fig. 15a and first stacked bar in b) and smallest for the *ozone* experiment (purple bars in a and fifth stacked bar in b), consistent with Fig. 2. For CRE, the magnitude of the changes in the *cloud* experiment is even comparable to that found for the ScenarioMIP range (black bars in a). Based on previous studies using NorESM2 and CESM, large changes in the Arctic cloud feedback due to the cloud phase changes applied in *cloud* are expected (Tan and Storelvmo, 2019; Shaw et al., 2022). While the total additional warming in the





*snow* experiment is only slightly larger than in the *ozone* experiment (0.51 K vs 0.17 K, see Fig. 2e and f), the *snow* experiment (orange bars in a and fourth stacked bar in b) exhibits changes in HFLUX and HSTORE of comparable magnitude as that seen in *eddy* (blue bars on a and second stacked bar in b) and *iceSheet* (green bars in a and third stacked bar in b).

The ensemble spread for the NEEMS experiments is particularly large for certain seasons and regions. For Arctic surface temperature, the spread is small during summer and fall, particularly between September and mid-November, but increases 540 considerably during the cold months (Fig. 10). The sea-ice retreat regions, especially the Greenland-Barents Seas and the Beaufort-Chukchi Seas, exhibit the strongest ensemble-mean warming (Fig. 14a). These regions also experience the largest SAF during summer (Fig. S6a), the greatest ocean heat release to the atmosphere during winter (Fig. 13e), and an increase in low-level cloud cover (Fig. 14k). However, while the mean warming is large in all the sea-ice retreat regions (Fig. 14a), the ensemble spread is dominated by the variability on the Atlantic side, particularly in the Greenland-Barents Seas (Fig. 14b). 545 This is not only seen for the spread for the surface temperature field (Fig. 14b), but also for the spread in turbulent fluxes of latent heat (Fig. 14d), low-level cloud cover (Fig. 14l), CRE (Fig. 13b), HSTORE (Fig. 13f), HFLUX (Fig. 13h).

As mentioned earlier, the flow of energy between the ocean and atmosphere is greatly impacted by sea-ice retreat, low-level humidity, and clouds. The mechanisms causing the spread can operate locally, as small variations in sea-ice retreat among the ensemble members can have a profound impact by slightly modifying this flow and thus cause a spread in warming.

### 4.6 Emergent constraint of Arctic winter warming

The strong spatial resemblance between the future response in winter Arctic warming and the temperature contribution due to the surface heat flux shown in Fig. 13a and g is further corroborated by Fig. 16, which reveals a near-linear relationship between the projections of winter warming and winter ocean heat flux (Fig. 16a). This is not only seen for the sensitivity experiments (colored dots), but also holds when considering an ensemble of CMIP6 models (including NorESM2-LM; plus 555 signs and asterisk) with a correlation coefficient of 0.97 in both cases.

Furthermore, the future response in ocean heat flux is related to the loss in NH sea-ice area (Fig. 16b and Fig. 13e). This relationship is supported by the fact that future changes in the ocean heat flux (i.e., increase in ocean heat flux to the atmosphere) are predominantly happening in the regions where sea-ice has retreated (Fig. S5 and S8).

Finally, we use the concept of emergent constraints to attain an estimate of the future Arctic winter warming. With this 560 approach, a simulated future change can be constrained by observations if a strong linear relationship exists between it and the simulated and observable historical metrics (Bourgeois et al., 2022). Figure 16c reveals that the spread in the future changes in winter Arctic surface temperature is strongly correlated with the simulated historical trend in NH winter sea-ice area, both when considering the sensitivity experiments and when considering the CMIP6 models (correlation coefficients of −0.78 and −0.79, respectively). This relationship demonstrates that models or experiments that simulate strong contemporary winter 565 sea-ice decline also project strong winter Arctic warming in the future, and vice versa for those that simulate weak sea-ice decline. Applying this relationship, we can calculate a constrained estimate of future Arctic winter warming by normalizing the product of the conditional probability density function of the emergent relationship and the probability density function of the observational constraint, that is, the contemporary sea-ice trend for years 1979–2017 (red vertical dashed line and





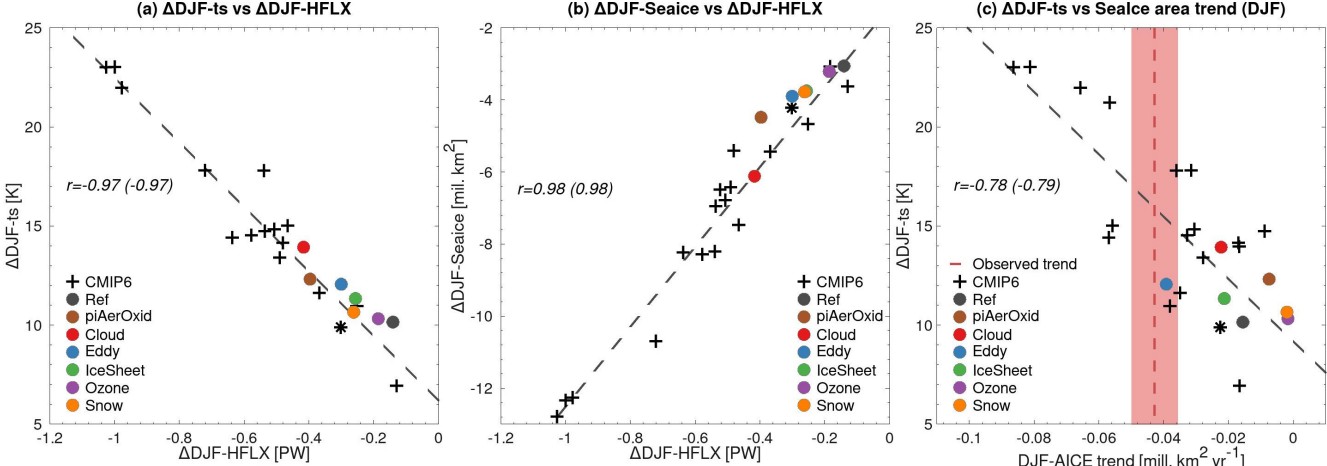

**Figure 16.** Scatter plots of the future response (Table 2) in Arctic winter surface temperature versus the response in winter ocean heat flux (a), the future response in winter NH sea-ice area versus the response in winter ocean heat flux (b), and the future response in Arctic winter surface temperature versus the contemporary (1979–2017) trend in winter NH sea-ice area (c). Values from the baseline (Ref) and sensitivity runs are shown as colored circles; additionally, to facilitate comparison with other models, we also show corresponding values from NorESM2-LM as asterisk and from other selected CMIP6 models (Supplementary Table S1) as plus signs. The vertical red dashed line and shading in c depict the observed sea-ice trend and uncertainties from Cavalieri and Parkinson (2012). The numbers in each panel represent the correlation coefficients (r) for the CMIP6 models and for both the CMIP6 models and all NorESM2-MM sensitivity experiments presented in this study (the latter given in parenthesis).

shading; Cavalieri and Parkinson, 2012). For more details, the readers are referred to Bourgeois et al. (2022). The resulting

observationally constrained estimate of future Arctic winter warming is 14.4±4.0 K. As demonstrated in Sect. 4.3, the projected loss of Arctic sea-ice in our sensitivity experiments strongly depends on its initial state during the historical period. This reiterates the importance of capturing contemporary sea-ice dynamics in order to increase the accuracy of the projected polar amplification (i.e., future Arctic warming).

## 5  Summary and discussion

While it is well known that the Arctic is warming faster than other regions on Earth, future projections are characterized by particularly large uncertainty for this region. To better our understanding of the sensitivity of Arctic amplification, we have carried out a coordinated set of fully coupled historical and future scenario experiments with NorESM2 with various changes to processes that are important for the projected surface energy budget in the Arctic: updated representation of mixed-phase clouds (*cloud*), updated eddy processes in the upper ocean (*eddy*), improved snow (on sea ice) processes (*snow*), interactive

ozone chemistry (*ozone*), and having an *interactive* Greenland ice-sheet model (*iceSheet*). Using data from these experiments,





we assess how these modifications affect the future Arctic climate, particularly during winter, when the projected warming is largest.

We find that all modifications (*cloud*, *eddy*, *iceSheet*, *ozone*, and *snow*) are associated with *additional* future Arctic warming relative to the future warming seen in the CMIP6 baseline experiments (historical and ssp585). The magnitude of the warming varies substantially between the experiments, with the most extreme warming found in *cloud* and the smallest in *ozone*. By the end of the 21st century, the range of warming between *cloud* and *ozone* reaches 8.6 K, which is almost as large as the ScenarioMIP range of 10.1 K (computed as the difference between ssp126 and ssp585).

The surface warming signal is largely captured by its first EOF, which displays relatively spatially uniform Arctic warming that starts to equilibrate between 2100 and 2200. The secondary temporal variations are described by the second EOF which is characterized by strongest warming rate in the Barents-Kara Seas and over and adjacent to the Chukchi Sea, and cooling over and south of Greenland which peak around 2100 and decline thereafter. Warming towards equilibration is enhanced in the sensitivity experiments, particularly in *cloud*, whereas the regional differences in warming rate vary between experiments.

The warming is not limited to the surface, but also extends into the upper ocean with the maximum warming occurring around 300–400 m depth. Like in the atmosphere, the *cloud* experiments exhibit the strongest warming. The *iceSheet* experiments warm almost as much as *cloud*, while the *eddy* experiments undergo very little warming. The relatively large ocean warming in *iceSheet* is not associated with a particularly warm future state, but with the historical experiment having a weak Atlantic Water signature, that is, being colder than the other experiments. More work is needed to understand what is causing this, but it could be related to changes in the runoff from the ice sheet.

The sea ice also responds to the warming, but the magnitude of the future response is highly dependent on the present-day state. While the present-day sea-ice area and volume is already overestimated in the baseline experiments, it is even thicker and more extensive in the historical *eddy*, *iceSheet*, *ozone*, and *cloud* experiments, resulting in a stronger future reduction in sea-ice volume than in the baseline. The *snow* experiment on the other hand has the most realistic sea-ice area and extent, resulting in weaker future sea-ice loss compared to the baseline, and ice-free summer conditions ahead of all other experiments.

Decomposing the surface temperature response into contributions from different processes shows that the winter warming is associated with changes in clouds at all levels (CRE), ocean heat storage (HSTORE), and increased longwave downwelling radiation from the lower atmosphere (LWCS). The warming is counteracted by the temperature contributions associated with turbulent fluxes (HFLUX), which largely mirror HSTORE and represent a flux of energy from the ocean to the lower atmosphere. All experiments exhibit the largest regional changes in the sea-ice retreat regions, where heat absorbed by the ocean during summer is released into the lower atmosphere through turbulent fluxes. This process increases low-level cloud cover and near-surface specific humidity, which subsequently enhances surface warming by increasing downwelling longwave radiation. While these changes are seen in all the sea-ice retreat regions, the ones on the Atlantic side are characterized by the largest spread among the NEEMS experiments.

Finally, we show that the spread in future Arctic surface warming simulated in the different experiments is strongly correlated with projected changes in the air-sea heat fluxes, which in turn are determined by the future NH sea-ice loss. This relationship is found to be robust also for an ensemble of CMIP6 models. Applying an emergent constraint, we find that the large uncertainty




in the projected Arctic surface warming during winter can be constrained by the contemporary trend in the NH sea-ice area. Despite the non-linear feedback and complex interplay between the different climate components, this relationship suggests that accurately simulating the contemporary sea-ice dynamics is key for constraining uncertainties in the future projected Arctic warming. We note that our emergent constraint works for the ssp585 scenario, as all our sensitivity experiments are based on this scenario. Future studies considering other scenarios would be valuable to assess the robustness of this relationship.

Ocean heat transport is instrumental in sea-ice retreat. In the Greenland-Barents Seas region, where the ensemble spread in the NEEMS experiments is particularly large, the ocean heat transport is relatively small (0.02 PW NEEMS ensemble mean), but the ensemble range is from -0.03 PW in the CMIP6 baseline simulation to 0.05 PW in *piAerOxid*. Further investigation is needed to better understand the complex interplay between the Arctic ocean warming, the change in ocean heat transport, and the corresponding sea-ice response. Furthermore, the details of ocean warming, particularly whether the warming is confined to the surface or extends more into the subsurface, have significant implications for cloud responses (Gjermundsen et al., 2021). The interactions between ocean warming and cloud changes in the Arctic require more examination.

The impact of increasing model resolution on the simulated Arctic Ocean remains unexplored in this work. Some model-dependent results show that increasing model resolution generally improves key features and process of ocean circulation in the North Atlantic and Arctic Ocean (Marzocchi et al., 2015), but challenges remain for an overestimated deep-water formation in the subpolar North Atlantic (Koenigk et al., 2021). Deep-water formation is also expected to increase under a future warming climate with sea-ice retreat in the Arctic Ocean (Bretones et al., 2022), and eddy-rich models may be essential for modelling the northward extension of heat and salt transport to the Arctic Ocean (Moreno-Chamarro et al., 2021).

Our results show that dynamic coupling with the Greenland ice sheet largely has a weak impact on the fields investigated. This is in line with recent results from Haubner et al. (2025), who, based on the same *iceSheet* experiments that were considered here, showed that the impact of this coupling on the atmosphere is small over the historical and future periods up until approximately year 2100 because the effect of ice-sheet topography changes remains minor in comparison to the dominant GHG-induced warming. After 2100, elevation changes become large enough to cause appreciable warming at the regional scale. With respect to ocean impacts, they find changes in the freshwater flux to have a limited influence on the ocean over the historical period, where ice-sheet changes are rather small; however, as mass loss from the ice sheet starts to increase in the 21st century, the freshwater flux causes some changes in the stratification and circulation around Greenland.

Another model update that largely has a weak impact on the results is the introduction of a chemistry scheme (TS1) to explicitly describe ozone and other chemical species in the *ozone* experiments. This could be due to the TS1 scheme being similar to the chemistry scheme in the CESM2-WACCM experiments that were used to create the ozone and oxidant climatologies for the baseline experiments, resulting in the distributions of ozone and secondary aerosols being similar in the baseline and *ozone* experiments, and hence underestimating the impact of interactive ozone chemistry. Comparison with sensitivity experiments using climatologies taken from other models could help shed light on this, but this is beyond the scope of this paper.

Despite decades of progress in Earth system modeling, our understanding of the coupled interactions between various climate components and how they impact future Arctic changes remains incomplete. Several key processes have been identified and further investigated in this manuscript, while others remain unexplored and uncertain. For instance, the role of ocean bio-



**Table 3.** Data overview for NorESM2-MM experiments used in this paper. CMIP6 data can be retrieved from the Earth System Grid Federation (ESGF) and NEEMS (NorESM2 Ensemble Exploring Model Sensitivity) and other data from the Norwegian Infrastructure for Research Data (NIRD) Research Data Archive (RDA). Note that archiving of the cloud, snow, ozone, and piAerOxid experiments are currently in progress.

| Experiment | Dataset | Archive | Data citation | DOI |
|---|---|---|---|---|
| historical | CMIP6 | ESGF | Bentsen et al. (2019a) | 10.22033/ESGF/CMIP6.8040 |
| ssp585 | CMIP6 | ESGF | Bentsen et al. (2019e) | 10.22033/ESGF/CMIP6.8321 |
| ssp370 | CMIP6 | ESGF | Bentsen et al. (2019c) | 10.22033/ESGF/CMIP6.8270 |
| ssp245 | CMIP6 | ESGF | Bentsen et al. (2019d) | 10.22033/ESGF/CMIP6.8255 |
| ssp126 | CMIP6 | ESGF | Bentsen et al. (2019b) | 10.22033/ESGF/CMIP6.8250 |
| hist-cloud | NEEMS | NIRD RDA | – | – |
| ssp585-cloud | NEEMS | NIRD RDA | – | – |
| hist-eddy (1850–1899) | NEEMS | NIRD RDA | He et al. (2024a) | 10.11582/2024.00052 |
| hist-eddy (1900–2014) | NEEMS | NIRD RDA | He et al. (2024b) | 10.11582/2024.00054 |
| ssp585-eddy | NEEMS | NIRD RDA | He et al. (2024c) | 10.11582/2024.00051 |
| hist-iceSheet | NEEMS | NIRD RDA | Goelzer (2024b) | 10.11582/2024.00080 |
| ssp585-iceSheet | NEEMS | NIRD RDA | Goelzer (2024a) | 10.11582/2024.00079 |
| ssp585-iceSheet-ext | NEEMS | NIRD RDA | Goelzer (2024c) | 10.11582/2024.00081 |
| hist-snow | NEEMS | NIRD RDA | – | – |
| ssp585-snow | NEEMS | NIRD RDA | – | – |
| hist-ozone | NEEMS | NIRD RDA | – | – |
| ssp585-ozone | NEEMS | NIRD RDA | – | – |
| hist-piAerOxid | Other | NIRD RDA | – | – |
| ssp585-piAerOxid | Other | NIRD RDA | – | – |

geochemistry controls on the climate active trace gases in the atmosphere, such as DMS, could have considerable implications for polar amplification (Schwinger et al., 2017).

*Code and data availability.* The CMIP6 version of the NorESM2 code can be downloaded from Zenodo (https://doi.org/10.5281/zenodo.3905091; Seland et al., 2020a). NorESM is also freely available on GitHub (https://github.com/NorESMhub/NorESM). The data from the NorESM2-MM CMIP6 experiments are available through the Earth System Grid Federation (e.g., https://esgf-node.llnl.gov/search/cmip6/) and the data from the NorESM2-MM *eddy* and *iceSheet* experiments can be retrieved from NIRD Research Data Archive (https://archive.sigma2.no/); the *cloud*, *snow*, *ozone*, and *piAerOxid* experiments are in the process of being uploaded into the same archive. See Table 3 for a detailed overview.




*Author contributions.* LSG coordinated the paper in collaboration with JT and MS. LSG and JT designed the common sensitivity experi-
ments' protocol. Specific experiment design and execution was done by ØS and TS for *cloud*; AN and YH for *eddy*; HG, PML, and AB
for *iceSheet*; JBD for *snow*; DJLO for *ozone*, and ØS and DJLO for *piAerOxid*. The first draft of the abstract was written by LSG, the
introduction by LSG and JT, the model description (Sect. 2) by LSG, the description of *cloud* (Sect. 3.1) by TS and ØS, the description of
*eddy* (Sect. 3.2) by AN and YH, the description of *iceSheet* (Sect. 3.3) by HG, AB, and PML, the description of *snow* (Sect. 3.4) by JDB, the
description of *ozone* (Sect. 3.5) by DJLO, and the description of the other experiments (Sect. 3.6) by LSG. Analysis was performed by JT
and LSG for Sect. 4.1, CG and MB for Sect. 4.2, JBD for Sect. 4.3, TT with contributions from LSG for Sect. 4.4, AG and AR for Sect. 4.5,
and JT with contributions from DT for Sect. 4.6; SO and CL contributed to the text in all sections. The summary and discussion was written
by LSG, JT, AB, JBD, AG, HG, YH, PML, DJLO, and TT. All authors reviewed and contributed to the final version of the paper.

*Competing interests.* The authors declare that there are no competing interests.

*Acknowledgements.* All authors received funding from the Research Council of Norway (RCN) project KeyCLIM (project number 295046).
In addition, LSG was supported by PolarRES, a project funded under the European Union's Horizon 2020 research and innovation programme
(grant number 101003590), LSG, CL, and SO were supported by the RCN project BASIC (project number 325440), HG and PML were
supported by the RCN project GREASE (324639), AN was supported by the RCN project TopArctic (314826), AG was supported by the
European Research Council through grant 101081661, and TS was supported by the European Research Council through grant 101045273.
Storage resources were provided by Sigma2 — the National Infrastructure for High Performance Computing and Data Storage in Norway
(projects NN9252K, NS9034K, NN8006K, NS2345K, NS9560K, NS9252K, and NS8006K.) The production of the merged CryoSat-SMOS
sea ice thickness data was funded by the ESA project SMOS & CryoSat-2 Sea Ice Data Product Processing and Dissemination Service, and
data from 20101201 to 20220228 were obtained from AWI. We acknowledge Stefan Hofer for sharing source code modifications that were
used in the *cloud* experiments.



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
