# Peer review of "Sensitivity of winter Arctic amplification in NorESM2"

_EGUsphere, 2025_

## Author Comment (AC1)

**Response to RC1**

**Main comments:**

This manuscript explores the uncertainties in projections of Arctic amplification due to several key parameters that are known to be important contributors to uncertainties in climate projections. The structure and readability of the manuscript is very clear, and the description of the experiment set up is convincing. The only major concern I have is that the removal of a known bug influencing mixed-phase clouds is not really a sensitivity experiment. The bug should have been corrected, and another sensitivity experiment designed. In addition, the other sensitivity experiments should have been using the mixed-phase cloud bug fix.

*Response: Thank you very much for your comments. We are happy to hear that you find the structure and readability of the paper clear and the experiment description convincing. Regarding the cloud experiments, they actually involve multiple changes beyond the ice nucleation bug fix. Changes were made both to the efficiency of the Wegener-Bergeron-Findeisen process (liquid-to-ice conversion), the phase of detrained cloud water from convection and the fraction of dust and soot particles assumed to be able to nucleate ice, with the goal of obtaining a complex sensitivity simulation that is consistent with cloud phase vertical profiles as found in satellite retrievals. We will revise the text to place adequate emphasis on these parameterization changes and less on the bug fix.*

*We will clarify this by changing the description of the cloud experiments in the intro (paragraph starting on line 47) from:*

> *(...) "(1) improved representation of mixed-phase clouds, achieved by rectifying an error in the ice-crystal nucleation (Shaw et al., 2022; McGraw et al., 2023) and biases in the representation of cloud phase (Cesana et al., 2015)" (...)*

*to (new text shown in bold):*

> *(...) " (1) improved representation of mixed-phase clouds, achieved by rectifying **a bias in the** representation of cloud phase **relative to satellite observations** (Cesana et al., 2015)" (...)*

*We will also update second and third paragraphs in the description of the cloud experiments (starting on line 111) from:*

> *(...) "However, an overlooked limiter, which sets the maximum number of cloud ice particles, inadvertently negated this scheme's ability to produce changes in the ice crystal number due to the heterogeneous ice nucleation. This limiter has been replaced in cloud by one that simply ensures that the number of nucleated ice crystals does not exceed the number of available INPs (as originally intended), following Shaw et al. (2022) and McGraw et al. (2023).*

*However, with this correction alone, the fraction of cloud ice in the temperature range between −40 and 0∘ C increases considerably, and is no longer consistent with cloud phase observed by active remote sensing. This is problematic, because it is well established that cloud phase exerts a strong influence on the simulated extratropical cloud feedback and thus climate sensitivity (Tan et al., 2016). This issue is addressed in cloud by also reducing the efficiency of the Wegener-Bergeron-Findeisen process (i.e., the conversion from liquid to ice in mixed-phase conditions), adjusting the fraction of dust and black-carbon aerosols that are assumed to act as INPs, and changing the assumed thermodynamic phase of convective detrainment, such that cloud phase matches satellite observations both qualitatively and quantitatively (Hofer et al., 2024). "*

*to (new text shown in bold):*

*(...) "**This parameterization in turn strongly influences the** fraction of cloud ice in the temperature range between −40 and 0°C. **In a previous study using NorESM2, simulated cloud phase was found to be inconsistent** with cloud phase observed by active remote sensing **(Hofer et al., 2024)**. This is problematic, because it is well established that cloud phase exerts a strong influence on the simulated extratropical cloud feedback and thus climate sensitivity (Tan et al., 2016). **Hofer et al. (2024) found that clouds in NorESM2 contained too much ice and not enough liquid compared to the observations, despite the fact that the simulations in that study included a bug that severely limited heterogeneous nucleation. Correcting the bug and following the suggestions by Hofer et al. (2024) for how to adjust cloud phase in NorESM2, this** issue is addressed in cloud by reducing the efficiency of the Wegener-Bergeron-Findeisen process (i.e., the conversion from liquid to ice in mixed-phase conditions), **by** adjusting the fraction of dust and black-carbon aerosols that are assumed to act as INPs, and **by** changing the assumed thermodynamic phase of convective detrainment, such that cloud phase matches satellite observations both qualitatively and quantitatively (Hofer et al., 2024)."*

*We agree that all simulations should ideally have been run with the bug fix, but most of the simulations were conducted prior to the discovery of the bug and repeating them all with the bug fix is unfortunately not feasible.*

While most of the parts are very clearly written and presented, the result section including some of the figures needs some clarification / correction as detailed in the minor comments.

*Response: Thank you for pointing out places where there is need for corrections or clarifications in the results section and figures. We have described in detail how we will address this in the responses to the minor comments below.*

Furthermore, there are several relevant and recent papers that have not been cited in the manuscript (for example Cai et al., 2025; Douville, 2023). There also is an older paper by Bracegirdle and Stephensen (2012) that is relevant. Methodologies and results should be compared and contrasted to the present study.

*Response: Thanks for making us aware of these papers. Indeed, our emergent constraint result (Sect. 4.6) shares similar features with Cai et al. (2025) and Bracegirdle and Stephenson (2012). The following paragraph will be added at the end of Sect. 4.6:*

> *"We note that the relationship between the present-day Arctic sea-ice cover and future Arctic warming has been demonstrated for CMIP3 and CMIP5 models and more recently for CMIP6 ESMs (Bracegirdle and Stephenson, 2012; Cai et al., 2025) based on similar methods (emergent constraints), but applying slightly different variables (winter Arctic amplification versus present-day temperature in Bracegirdle and Stephenson, 2012; and annual-mean Arctic warming versus present-day global warming trend and the mean sea-ice cover in Cai et al., 2023). Our results complement their findings that in addition to the mean state, the historical transient change in the sea-ice cover could serve as an additional constraint, particularly for projections of Arctic winter warming."*

*Also, we will add the following sentence in the summary and discussion, after the sentence ending on line 616:*

> *"This is consistent with Douville (2023), who found that the observed recent trend in NH sea-ice extent can be used to constrain Arctic surface warming during the extended winter season using the Kriging for Climate Change constraint method."*

**Minor comments:**

Abstract, lines 14-16: Similar to results from Pithan et al.?

*Response: Do you mean Pithan and Mauritzen 2014? Then yes, this is similar to their results. It is also similar to the results in Boeke and Taylor (2018).*

*We will add these references to the sentences starting on 604 as follows (new text shown in bold):*

> *"Decomposing the surface temperature response into contributions from different processes shows that the winter warming is associated with changes in clouds at all levels (CRE), increased ocean heat storage (HSTORE) and longwave downwelling radiation from the lower atmosphere (LWCS)__, similar to in Boeke and Taylor (2018) and Pithan and Mauritsen (2014)__. The warming is counteracted by the temperature contributions associated with turbulent fluxes (HFLUX), which largely mirror HSTORE and represent a flux of energy from the ocean to the lower atmosphere __as in Boeke and Taylor (2018)__."*

Line 47: can the rectification of an error be declared as a model improvement (see also main comments)?

*Response: Please see the response to the first main comment.*

Lines 131/132: with the cloud physics changed, how can we know that we are still in a well-balanced state? Would there be a spin-up effect in the beginning of the historical *cloud* simulation?

*Response: The top-of-the-model radiative imbalance was found to be -0.048 W/m$^2$ in a 30 year long pre-industrial control cloud experiment starting from the same initial conditions as the historical simulation. This is slightly lower than the top-of-the-atmosphere radiative imbalance in the CMIP6 pre-industrial control experiment for NorESM2-MM, which is -0.065 W/m$^2$ (Seland et al. 2019).*

Line 257: after the 60 years, is the model spun up?

*Response: The 3-D concentration fields of the newly added tracers are expected to be well spun up after 15 to 20 years. These new tracers started from a zero concentration at the beginning of the spin-up and it takes on the order of a decade before stratospheric concentrations of NOx, ozone and some long-lived green-house gases (LLGHGs) such as $CH_4$, $N_2O$, CFC-11 and CFC-12 come into a steady state. The fact that the model has a relatively low model top and coarse vertical resolution in the stratosphere limits the need for a long spin-up. In a second phase, the new model should also come into equilibrium with respect to the slightly different forcings (different ozone, different aerosol, and different LLGHGs). As these forcings were relatively close to their original values, a 60-year spinup seemed reasonable. Nevertheless, there could still be small trends (e.g., a slight cooling from the higher sea-salt and dust concentrations) due to the long time-scale of the ocean to come into thermal equilibrium.*

Line 264: 3.7 or 7.0 W/m2?

*Response: It should be 7.0, this will be corrected.*

Lines 284/285: Are these numbers for 30-year periods or for last year minus pre-industrial (after having applied the running mean?), or something else? Please specify.

*Response: These numbers are for the year 2100 minus year 1850 after applying the 10-year running mean. We will clarify this in the revised text by changing the sentence from:*

*"By 2100, NorESM2-MM projects annual-mean Arctic surface warming ranging from 2.76 to 9.66 K under ssp126 and ssp585, relative to the pre-industrial state (Fig. 1a)."*

*to:*

*"In NorESM2-MM, the projected annual-mean Arctic surface warming in 2100 ranges from 2.76 to 9.66 K under ssp126 and ssp585, relative to the pre-industrial state of 1850 (values are calculated after applying a 10-year running mean; Fig. 1a)."*

Line 303: "more than exceeding": duplication.

*Response: We will remove "exceeding".*

Lines 334-339: the cold bias is large compared to the warming signal, resulting in a situation where the difference future simulations minus PHC3.0 is smaller than the difference PHC3.0 minus historical simulations in many cases (as shown in Fig. S2). This merits a mention to allow the reader to judge the magnitude of the cold bias.

*Response: This is indeed an interesting point, thanks for pointing it out. We will add the following sentence to the end of the paragraph starting at line 334:*

> *"It is not clear how the cold bias affects the future changes, but it should be noted that for most experiments, the magnitude of the cold bias in the historical runs exceeds half of the projected future warming (Fig. S2)."*

Line 380: "is know" -> "is known"

*Response: We will correct this.*

Lines 408/410: more Barents Sea than Kara Sea?

*Response: Agreed, we will mention the Barents Sea instead of the Kara Sea.*

Lines 418-420: The North Atlantic Warming Hole signature seems to be present in EOF2 of the NH ts while it is out of the area for the Arctic.

*Response: Agreed, thanks for pointing this out. We will change the sentence from:*

> *"Arctic EOF2-related warming is stronger than the baseline in some experiments (eddy, iceSheet, ozone), but weaker in others (cloud, snow), suggesting the latter exhibit a less prominent lag in North Atlantic warming (described as an "Atlantic warming hole", c.f. Kiel et al., 2020; He et al., 2022)."*

*to:*

> *"Arctic EOF2-related warming is stronger than the baseline in some experiments (eddy, iceSheet, ozone), but weaker in others (cloud, snow)"*

Line 486: "low-level cloud cover increases across all seasons": can't see this from Fig. 12b, and only partly from Fig. S7b, f, and j.

*Response: Thanks for pointing this out. We will add the following figure to the supplement (will be Fig. S8 in the revised version), which is similar to Fig. 12, but for the Arctic Ocean (80°N–90°N):*

[Figure]

*Figure RC1.1: Monthly evolution of the future response (Table 2) in total cloud cover (a), low-level cloud cover (b), medium-level cloud cover (c), and high-level cloud cover (d) for the Arctic Ocean (defined as 80°N–90°N) for the CMIP6 baseline (black line), cloud (red line), eddy (blue line), iceSheet (green line), snow (orange line), ozone (purple line), and piAerOxid (brown line). The annual, DJF, and JJA mean values, averaged across the baseline and five NEEMS experiments (cloud, eddy, iceSheet, snow, and ozone), are shown in the left legend. Units are in percentage change.*

*We will also change the sentence starting on line 485 from:*

> *"However, over the central Arctic Ocean, the low-level cloud cover increases across all seasons (Fig. 12b and Fig. S7b, f, and j)."*

*to:*

> *"However, over the central Arctic Ocean, the low-level cloud cover generally increases across all seasons (shown in Fig. S8 and in Fig. S7b, f, and j for DJF, May, and August)."*

Line 531: staced -> stacked

*Response: We will correct this.*

Line 570: I'm surprised that it is only 14.4 K and not around 16 K plus minus some error bar. I would have expected around 16 if looking at the crossing of the red and the black dashed lines in Fig. 16c. Here, at least a brief description of the method to determine the emergent constraint is required.

*Response: It was indeed quite interesting for us as well to see that the constrained value is similar, but with slightly smaller uncertainties than the unconstrained value. We will revise the following sentence (Sect 4.6):*

> *"The resulting observationally constrained estimate of future Arctic winter warming is 14.4±4.0 K"*

*to*

> *"The resulting observationally constrained estimate of future Arctic winter warming is 14.4±4.0 K, which is comparable to, but with slightly smaller uncertainty, than the unconstrained CMIP6 multi-model mean of 14.2±4.2 K."*

*Below, Fig. RC1.2 illustrates how the constrained value was estimated. The emergent constraint generates the probability density function P(y) of the constrained*

*warming rate (magenta curve in y-axis) by normalizing the product of the conditional pdf of the emergent relationship P(y|x) and the pdf of the observational constraint P(x) (red curve). A brief description of this has been provided on line 566-569 and we have cited Bourgeois et al. (2022), which provides a more detailed description of emergent constraint calculation.*

[Figure]

**Figure RC1.2:** *Similar to Fig 16c in the manuscript but depicting the gaussian distribution of the observational estimate of winter sea-ice trend (red line), and the unconstrained (blue curve) and constrained (magenta curve) winter temperature change.*

*The reason that the constrained value is not close to 16 K is because models that lie closer to the observational uncertainty also have lower warming rates than 16 K.*

Fig. 1: At first, I struggled to understand why there are coloured lines in panel a), and grey lines in panel c), given the different legends in panels a) and c). I think there would be space to have both legends in both panels. There are similar cases in other Figures as well.

*Response: We will update the legends in Fig. 1a,c, Fig. 6a,b and Fig. S1a,c so that all lines shown within the panels are defined in the legends as shown below. We will also update the figure captions where needed.*

[Figure]

**Figure RC1.3:** *Updated version of Fig. 1 with legends for all lines shown in a and c.*

[Figure]

**Figure RC1.4:.** *Updated version of Fig. 6 with legends for all lines shown in a and b.*

[Figure]

**Figure RC1.5:** *Updated version of Fig. S1 with legends for all lines shown in a.*

Fig. 3h: not sure what to make out of a negative range (between 65 and 70 N at around 700 to 800 m depth). Maybe better to say "difference" instead of "range", as it is SSP585 minus SSP126? Similar applies to Fig. 5h.

*Response: We prefer to keep the term "ScenarioMIP range", but we now clearly specify what is meant by it (ssp585-ssp126) in the captions of all relevant figures. In addition, the ScenarioMIP range is already defined in Table 2.*

*The negative temperature difference between 65 and 70°N at around 700 to 800 m depth coincides with a positive ideal age difference of 10–30 years (Fig. RC1.6, left panel), meaning that this region in ssp585 is less ventilated than ssp126. The ssp585 is also fresher in this region compared to ssp126 and even fresher towards the surface (right panel). Thus, we expect that the ssp126 simulation is able to ventilate this region with more saline and warmer water compared to ssp585, leading to the negative temperature difference.*

[Figure]

**Figure RC1.6:** *Future changes in zonal-mean Arctic ideal age (left panel) and salinity (right panel) during winter for the ScenarioMIP range (ssp585–ssp126).*

Figs. 5 and S4: in principle you are showing the sea ice thickness averaged over each model grid cell. Meter per unit area and calling this a volume is a bit confusing. Furthermore, it is difficult to judge the meaning of panel (a) without the simulated distribution of sea ice thickness in the historical period. Is practically all sea ice gone or is there still a substantial amount left during the last 30 years?

*Response: We agree that the "meter per unit area" is misleading and will change this to "volume per unit area (m)" to be consistent with the definition used in the sea ice model.*

*We also suggest including the historical sea ice volume in Figure S4 such that it shows the satellite observations, the baseline (historical) distribution, together with the biases for the baseline and all the sensitivity experiments.*

*We will update the figures as shown below:*

[Figure]

*Figure RC1.7: As Fig. 5, but with units in sea-ice volume per area.*

[Figure]

*Figure RC1.7: Updated version of Fig. S4, but with the baseline sea-ice volume added (panel b in the new version) and units in sea-ice volume per area.*

Fig. 6, caption, third last line: "… transition to and seasonal …" -> "… transition to a seasonal …"

*Response: We will update the sentence to:*

*"Summer sea-ice area (September) is shown to illustrate the transition to a seasonal Arctic ice cover, while March sea-ice volume represents the seasonal sea-ice maximum."*

Fig. 6, caption, last line: "Units are in mill km2": only valid for panel a).

*Response: Will change from "Units are in mill $km^2$." to "Units are mill $km^2$ (a) and thousand $km^3$ (b).*

Fig. 7, caption: check references to the panels.

*Response: The panel references will be corrected.*

Fig. 9, caption, line 6: even the historical period is characterized by small trends. For computation of the variance, has a detrending been applied?

*Response: A detrending has indeed been applied. We will specify this in the caption.*

Fig. 14, caption, line 1: fluxe -> flux

*Response: We will correct this.*

Fig. S3: noteworthy AMOC strength peak in all simulations but hist/ssp585-piAerOxid. How about commenting on this?

*Response: We agree that this is an interesting point. We will add the following between the sentences ending and starting on line 329:*

*"Changes in AMOC strength can on the other hand be linked to the future changes in piAerOxid. This experiment exhibits a large sub-surface warming (Fig. 3g) and also has the smallest AMOC reduction over the 21st century, largely because it is the only experiment which does not exhibit a maximum toward the end of the 20th century. This is in line with (Hassan et al., 2021), who found robust AMOC weakening in response to decreased anthropogenic aerosols in CMIP6."*

Fig. S7, caption: August and May swapped

*Response: We will correct this.*

References:

Bracegirdle and Stephensen, 2012: On the Robustness of Emergent Constraints Used in Multimodel Climate Change Projections of Arctic Warming in: Journal of Climate Volume 26 Issue 2 (2012)

Cai et al., 2025: Lessened projections of Arctic warming and wetting after correcting for model errors in global warming and sea ice cover | Science Advances

Douville, 2023: Robust and perfectible constraints on human-induced Arctic amplification | Communications Earth & Environment

---

## Author Comment (AC2)

**Response to RC2**

This manuscript clearly presents the experimental setup and effectively addresses key parameters contributing to uncertainties in climate projections. The methods are clearly described and outlined to the reader. In my opinion, the removal of a known bug affecting mixed-phase clouds is more of an update and not an independent experiment.

*Response: Thank you very much for your comments. We are happy to hear that you find the manuscript to be clear and effectively address key parameters contributing to uncertainties in climate projections. Regarding the cloud experiment, it actually involves multiple changes beyond the ice nucleation bug fix. Changes were made both to the efficiency of the Wegener-Bergeron-Findeisen process (liquid-to-ice conversion), to the phase of detrained cloud water from convection, and to the fraction of dust and soot particles assumed to be able to nucleate ice, with the goal of obtaining a control simulation that is consistent with cloud phase based on satellite retrievals. We will revise the text describing the cloud experiments in the introduction and in section 3.1 to place more emphasis on these parameterization changes and less on the bug fix. Please see the response to the first main comment from reviewer 1 for a detailed overview of the suggested changes to the text.*

Enclosed below you can find some minor technical comments regarding the figures:

Fig. 5: Include units in the colorbar within the figure.

*Response: We will change the figure as follows:*

[Figure]

***Figure RC2.1:*** *As Fig. 5, but with units shown below the color bar.*

Fig. 7: Indicate lack of units in the colorbar with something like [-]

*Response: We will add "unitless" to the colorbar title to indicate the lack of units.*

Fig. 13:  Include units in the colorbar within the figure.

*Response: We will add units to the colorbars as shown below.*

[Figure]

*Figure RC2.2: As Fig. 13, but with units shown below the colorbars.*